# Extrapolating from Regularised Solutions for Solving Ill-Conditioned Linear Systems in Machine Learning

**Disha Hegde**                                                    *d.hegde@soton.ac.uk*
*University of Southampton, UK*

**Jon Cockayne**                                                *jon.cockayne@soton.ac.uk*
*University of Southampton, UK*

**Chris. J. Oates**                                         *chris.oates@newcastle.ac.uk*
*Newcastle University, UK*
*The Alan Turing Institute, UK*

**Reviewed on OpenReview:** *https://openreview.net/forum?id=fqbkenUpRa*

## Abstract

Rapid prototyping of algorithms is a critical step in modern machine learning. Most algorithms exploit linear algebra, creating a need for lightweight numerical routines which – while potentially sub-optimal for the task at hand – can be rapidly implemented. For the numerical solution of ill-conditioned linear systems of equations, the standard solution for prototyping is Tikhonov-regularised inversion using a nugget. However, selection of the size of nugget is often difficult, and the use of data-adaptive procedures precludes automatic differentiation, introducing instabilities into end-to-end training. Further, while data-adaptive procedures perform multiple linear solves to select the size of nugget, only the result of one such solve is returned, which we argue is wasteful. This paper aims to circumvent the above difficulties, presenting `autonugget`; a `Python` package for automatic and stable numerical solution of linear systems suitable for rapid prototyping, and fully compatible with automatic differentiation using `JAX`. `autonugget` combines multiple linear solves using Richardson extrapolation to determine the solution of the ill-conditioned system, improving in accuracy over approximations based on a single nugget.

## 1 Introduction

Consider the problem of numerically solving $Ax = b$ for $x \in \mathbb{R}^d$. It will be assumed that $A$ is a symmetric positive definite matrix, but that $A$ is (possibly severely) ill-conditioned and, as a result, solving the linear system with standard methods is challenging. This setting is ubiquitous in applied machine learning like Bayesian optimisation (Garnett, 2023), emulation (Gramacy, 2020), and probabilistic numerics (Hennig et al., 2022).

In machine learning, a widely-used approach to improving the conditioning of the system so that it can be solved is to use *Tikhonov regularisation:*

$$x_\sigma := (A + \sigma I)^{-1}b, \qquad \sigma \geq 0, \tag{1}$$

where $x_\sigma \to x$ in exact arithmetic as the regularisation parameter $\sigma \to 0$. The regularisation parameter $\sigma$ is referred to as a *nugget*. For sufficiently regular matrices $A$, such as positive semi-definite matrices, the conditioning of $A + \sigma I$ improves monotonically as $\sigma$ is increased. However, in practice this often results in a conflation of two ideals:

1. **Regularisation:** as a technique for penalising large values in the solution vector, improving generalisation performance. In this setting the focus is on $\sigma > 0$.

2. **Ill-conditioning:** as a technique for improving the numerical performance of solver routines. In this setting the quantity of interest is $\sigma = 0$, but the ill-conditioning of $A$ causes numerical problems in the computational pipeline.

In this paper, the focus is solely on the latter: we assume that the nugget is being used only as a tool to improve the conditioning of $A$.

When using Tikhonov regularisation to improve conditioning, one usually takes $\sigma$ to be just large enough that a numerical approximation $\hat{x}_\sigma$ to $x_\sigma$ can be obtained to a required level of precision using a direct method (e.g. LU-factorisation). The popularity of Tikhonov-regularisation for prototyping of algorithms is due to the simplicity with which it can be implemented, and the presence of only a single degree of freedom to be selected. For subsequent code optimisation, one would ideally upgrade from a Tikhonov-regularised direct method to a more scalable or sophisticated linear algebra routine, such as a preconditioned iterative method (Trefethen and Bau, 2022), though we note that Tikhonov regularisation often forms a component of these routines, e.g. in constructing preconditioners (Cutajar et al., 2016) and in certain Krylov subspace methods (Gazzola et al., 2015). Moreover, while ideally the numerics should be improved to address instabilities resulting from poor conditioning, how often this actually happens could be questioned.

The aim of this paper is to address three key issues that arise in applications of Tikhonov regularisation to address poor conditioning:

1. **Size of the nugget:** The critical minimum value $\sigma_\star$ of $\sigma$ at which $x_\sigma$ can be stably computed (i.e. as $\hat{x}_\sigma$) will be unknown in general, and will depend sensitively on the actual values in $A$ and $b$.

2. **End-to-end training:** Data-adaptive choices of $\sigma$, for example based on the condition number of $A + \sigma I$ remaining below a specified threshold, are typically incompatible with automatic differentiation with respect to any variables $\theta$ contained in $A \equiv A_\theta$ and $b \equiv b_\theta$. This precludes straight-forward end-to-end training of machine learning algorithms via gradient descent.

3. **Data-efficiency:** While data-adaptive procedures perform multiple linear solves to select the size of nugget, only the result of one such solve is returned, which we argue is wasteful.

To resolve these difficulties we present `autonugget`; a `Python` package for hassle-free stable numerical solution of linear systems, fully compatible with automatic differentiation using `JAX` (Bradbury et al., 2018). To address data-efficiency, we explore the benefit of combining multiple linear solves corresponding to nuggets $\Sigma = \{\sigma_i\}_{i=1}^n$ using Richardson extrapolation, which enables `autonugget` to improve in accuracy over approximations based on a single nugget. Automatic selection of $\Sigma$ is performed by letting $\Sigma = h\Sigma_{\text{ref}}$ for some reference set $\Sigma_{\text{ref}}$ and selecting $h$ to (approximately) balance the error due to extrapolation with the error due to numerical solution of (1). The `autonugget` package is described in Section 3 and illustrated in Figure 1. Empirical evidence in support is presented in Section 4.

## 1.1 Related Work

The fundamental importance of linear algebra ensures (1) has been theoretically and empirically well-studied. On the other hand, we argue that there is an important gap at the interface of theory and practice with respect to existing work:

**Machine Learning** The role of $\sigma$ is often conflated in applied machine learning, on the one hand being a numerical parameter that one would ideally want to be small, while on the other hand serving as a 'regulariser' that affects downstream performance on a particular task. From this perspective, $\sigma$ is often folded into the hyper-parameters of a machine learning algorithm and tuned end-to-end (e.g., in Bayesian inference through marginalisation). Similarly, amortised approaches, which aim to learn a mapping from the inputs of the machine learning task to an appropriate value for $\sigma$, have been pursued (Chung and Español, 2017; Alberti et al., 2021; De Vito et al., 2022). However, both approaches represent a (possibly substantial) increase in the overall training cost. Further, there are important situations where one seeks the exact solution $x$ of (1) as $\sigma \to 0$; for instance in regression applications, where Tikhonov regularisation can lead to predictions

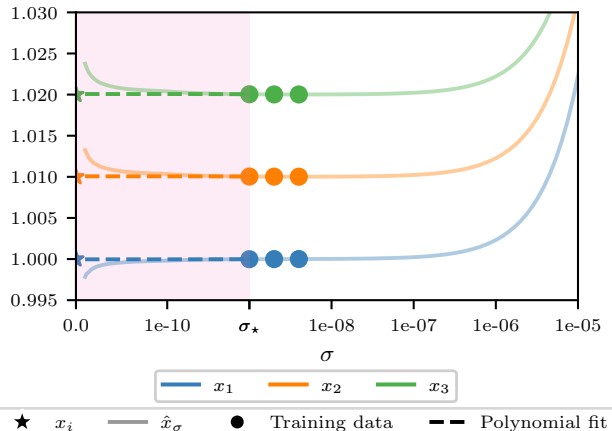

Figure 1: Illustration of `autonugget`: Here we plot numerical approximations $\hat{x}_\sigma$ to $x_\sigma$ as the nugget $\sigma$ is varied (to aid in visualisation, only the first three coordinates are displayed). The numerical approximations were obtained using `linalg.solve` in `numpy` (Harris et al., 2020), a direct method based on LU factorisation. There is a critical value $\sigma_\star$ (shaded) below which $\hat{x}_\sigma$ fails to reliably approximate $x_\sigma$. On the other hand, reliable calculations can be performed when $\sigma \geq \sigma_\star$, and extrapolation of these reliable data can even provide a better approximation to $x$ compared to $\hat{x}_{\sigma_\star}$.

that are under-confident (Andrianakis and Challenor, 2012). Guidance on the practical choice of $\sigma$ for exact solution is then limited; in our experience, manual choices based on the (in)sensitivity of the downstream output are often used. As a data-adaptive alternative, one can select $\sigma$ large enough that the condition number of $A + \sigma I$ remains below a specified threshold[1]; however, the introduction of conditional expressions can introduce an incompatibility with automatic differentiation, which can preclude gradient-based training. On a different note, the potential to accelerate computation in machine learning using the classical idea of Richardson extrapolation is receiving renewed attention (Bach, 2021; Oates et al., 2025) and its potential to accelerate linear algebra computations in modern machine learning has yet to be explored.

**Numerical Analysis** Numerical linear algebra is well-developed and the concepts that we discuss in this work are not novel. For instance, the theoretical application of Richardson extrapolation in the setting of Tikhonov regularisation was discussed as far back as Groetsch and King (1979); Thomas King and Chillingworth (1979). Theoretical insight into the data-adaptive selection of $\sigma$ can be found in the works of Gfrerer (1987); Engl (1987); Liu (2013). The topic has fallen out of fashion in numerical analysis as attention has turned to designing bespoke algorithms for particular problems, often with applications to partial differential equations in mind (Mardal and Winther, 2011; Kunoth et al., 2018). However, the sustained popularity of Tikhonov regularisation as a prototyping tool in modern machine learning suggests the time is right to return to this subject. In particular, there is a strong demand for a differentiable software implementation that provides the exact solution of (1) as $\sigma \to 0$, which the numerical analysis community have to-date not provided.

## 2 Methodology

This section presents `autonugget`. The premise is that there is typically a critical value $\sigma_\star$ such that the numerical approximation $\hat{x}_\sigma$ to $x_\sigma$ fails to be reliable for all $\sigma < \sigma_\star$. If one was able to estimate $\sigma_\star$, then one could generate data $\hat{x}_\sigma$ from the regime $\sigma \geq \sigma_\star$ and attempt to extrapolate these reliable data to predict the $\sigma \to 0$ limit (c.f. Figure 1). This amounts to an application of Richardson extrapolation, as explained

---

[1]The *condition number* $\kappa(A)$ of a non-singular matrix $A$ is defined as the ratio $\lambda_{\max}(A)/\lambda_{\min}(A)$ of the largest and smallest eigenvalues of $A$; a smaller condition number is often associated with improved accuracy and/or faster convergence of numerical methods (Trefethen and Bau, 2022).

in Section 2.1, and we prove that this enables convergence acceleration in Section 2.2. The problem of identifying $\sigma_\star$ is deferred to Section 2.3.

**Notation** Let $x \in \mathbb{R}^d$ and $A \in \mathbb{R}^{d \times d}$. The following norms will be used: $\|x\|_\infty := \max\{|x_i|\}_{i=1}^d$, $\|x\|_2 := (\sum x_i^2)^{1/2}$, and $\|A\|_{\mathrm{op}} := \sup_{\|x\|_2=1} \|Ax\|_2$. The minimum and maximum eigenvalues of $A$ are denoted $\lambda_{\min}(A)$ and $\lambda_{\max}(A)$, and the condition number is denoted $\kappa(A) := \lambda_{\max}(A)/\lambda_{\min}(A)$. The set of real-valued functions on a set $S \subset \mathbb{R}$ whose derivatives up to order $r$ are continuous is denoted $C^r(S, \mathbb{R})$. For $f \in C^r(S, \mathbb{R})$ and $S \subset \mathbb{R}$ bounded, let $\|f\|_\infty := \sup_{s \in S} |f(s)|$ and $\|f\|_{\infty, \Sigma} := \sup\{f(\sigma)\}_{\sigma \in \Sigma}$ for $\Sigma \subset S$. Further, for a linear functional $\Pi$ on $C^0(S, \mathbb{R})$, the operator norm

$$\|\Pi\|_{\mathrm{op}} := \sup_{0 \neq f \in C^0(S, \mathbb{R})} \frac{|\Pi(f)|}{\|f\|_\infty} \tag{2}$$

will be used.

## 2.1 Extrapolation Estimator

Richardson extrapolation (Richardson, 1911) is a classical idea from numerical analysis, whose potential in modern machine learning and statistics is only beginning to be appreciated (Bach, 2021; Oates et al., 2025). In brief, the idea is to construct polynomial approximations of the coordinate maps $\sigma \mapsto [x_\sigma]_i$ and to extrapolate the fitted polynomial to $\sigma = 0$. Since in practice we only have access to numerical approximations $\hat{x}_\sigma$, we aim to work with data for which $\sigma \geq \sigma_{\min}$ so that we can substitute $x_\sigma$ with a reliable approximation $\hat{x}_\sigma$ produced using a direct method.

Let $\mathcal{P} := \mathrm{span}\{p_1, \ldots, p_m\}$ where each $p_i : [0, \infty) \to \mathbb{R}$ is a polynomial. For the purposes of this paper the constant function is always an element of $\mathcal{P}$. Given a set $\Sigma = \{\sigma_i\}_{i=1}^n$, the *Vandermonde matrix* is denoted

$$\mathbf{V}(\Sigma) := \begin{bmatrix} p_1(\sigma_1) & \ldots & p_m(\sigma_1) \\ \vdots & & \vdots \\ p_1(\sigma_n) & \ldots & p_m(\sigma_n) \end{bmatrix}.$$

In the case where $m$ and $n$ are equal, the *Vandermonde determinant* is denoted $\mathrm{VDM}(\Sigma) = \det(\mathbf{V}(\Sigma))$. The set $\Sigma$ is called $\mathcal{P}$-*unisolvent* if $\mathrm{VDM}(\Sigma) \neq 0$. Assuming $\Sigma$ is $\mathcal{P}$-*unisolvent*, we can approximate $x$ using polynomial extrapolation applied coordinatewise to the training dataset.

**Definition 1** (Extrapolation Estimator). *Let $f : [0, \infty) \to \mathbb{R}^d$. Let $f_n \in \mathcal{P}^d$ interpolate $f$ (coordinatewise) on $\Sigma = \{\sigma_i\}_{i=1}^n$. Then $f_n(0)$ is called an* extrapolation estimator *of $f(0) \in \mathbb{R}^d$.*

The extrapolation estimator $f_n(0)$ exists and is unique provided that $\Sigma$ is $\mathcal{P}$-unisolvent. In this paper $f(\sigma) = x_\sigma$. The idea in `autonugget` is to estimate $f(0)$ using an extrapolation approach. To accomplish this we distinguish between the extrapolator $f_n$, based on exact solutions to the linear system, $x_\sigma$, and the extrapolator $\hat{f}_n$ based on approximations $\hat{x}_\sigma$. We focus on determining a set $\Sigma$ which balances the tension between accuracy of the approximations $\hat{x}_\sigma$, which is improved for large $\sigma$, and accuracy of the extrapolator $f(0)$ which is improved for small $\sigma$. The computational cost of the extrapolation estimator is $n$ times that of a single linear solve, since polynomial interpolation requires negligible overhead. Typically $n$ will be small, e.g. $n \in \{2, 3, 4\}$, so that the overall computational cost of `autonugget` is of the same order as a single application of a direct method.

## 2.2 Analysis in Exact Arithmetic

To understand the benefit of extrapolation we begin by analysing the error of extrapolation applied to exact data $x_\sigma$; the case where numerical approximation of $x_\sigma$ is taken into account is deferred to Section 2.3.

Let $\ell_i(\cdot; \Sigma)$ denote the *Lagrange polynomials*, defined as the elements of $\mathcal{P}$ for which $\ell_i(\sigma_j; \Sigma) = \delta_{i,j}$. These can be computed as

$$\ell_i(\sigma; \Sigma) = \frac{\mathrm{VDM}(\{\sigma\} \cup (\Sigma \setminus \{\sigma_i\}))}{\mathrm{VDM}(\Sigma)}.$$

The *Lebesgue function* of $\Sigma$ is denoted

$$\lambda(\sigma; \Sigma) = \sum_{i=1}^{n} |\ell_i(\sigma; \Sigma)|. \tag{3}$$

The following result, which we prove in Section A.2, indicates the potential benefit from extrapolation in this context:

**Theorem 1** (Extrapolation error bound)**.** *Let $f(\sigma) = (A + \sigma I)^{-1}b$ where $A$ is a symmetric positive definite matrix. Let $\mathcal{P} = \text{span}\{1, \sigma, \ldots, \sigma^{n-1}\}$. Let $\Sigma_{\text{ref}} = \{\sigma_i\}_{i=1}^{n}$ be $\mathcal{P}$-unisolvent. Let $f_n^h \in \mathcal{P}^d$ interpolate $f$ (coordinatewise) on $\Sigma_h = \{h\sigma_i\}_{i=1}^{n}$ for $h \in (0, 1]$. Then*

$$\underbrace{\|f(0) - f_n^h(0)\|_\infty}_{\text{extrapolation error}} \leq \underbrace{(1 + \lambda(0; \Sigma_{\text{ref}}))\sigma_{\max}^n \lambda_{\min}(A)^{-(n+1)}\|b\|_2}_{\text{constant in } h} \underbrace{h^n}_{\text{acceleration}},$$

*where $\sigma_{\max} := \max\{\sigma_i\}_{i=1}^{n}$.*

To interpret Theorem 1 one should consider $h$ to be varying and $n$ to be fixed. The extrapolation error is then seen to converge at a rate $O(h^n)$, where the $n$ nuggets that are used to generate the dataset are each of size $O(h)$. As a sanity check, for the case of a single linear solve and no extrapolation, the map $\sigma \mapsto x_\sigma$ is continuous and the error $x - x_{h\sigma}$ is indeed $O(h)$. Theorem 1 thus shows that the convergence rate (in $h$) is strictly improved when $n \geq 2$ data are used. There is of course also a trade-off, in terms of the $n$-dependent constants appearing in the bound[2] and in terms of the computational cost[3].

In practice however we do not have access to $x_\sigma$, only to a approximation $\hat{x}_\sigma$ obtained via the numerical solution of (1). Care is needed when using polynomial extrapolation applied to noisy data, because numerical errors can be amplified by higher-order terms in the polynomial. Understanding the trade-off between convergence acceleration and stability in finite precision arithmetic is the focus of Section 2.3.

## 2.3 Analysis in Finite Precision Arithmetic

To understand the practical performance of the extrapolation estimator we will characterise the estimator as a *projection* of the data-generating function, and then analyse how the projection is affected when the input data are corrupted. As before, we consider a polynomial basis $\mathcal{P}$ and a collection $\Sigma = \{\sigma_i\}_{i=1}^{n}$. Denote the projection

$$\Pi_\Sigma : C^0([0, \sigma_{\max}], \mathbb{R}^d) \to \mathbb{R}^d$$
$$f \mapsto f_n(0),$$

where $f_n(0)$ is the extrapolation estimator of $f(0)$ based on $\mathcal{P}$ and $\Sigma$ (cf. Definition 1). Note that $\Pi_\Sigma[f]$ can be defined for *any* function $f$, including $\hat{f}$, but one cannot hope to extrapolate a function that is discontinuous at $0$.

Recall that we cannot exactly evaluate $f(\sigma) = x_\sigma$, but rather we obtain $\hat{f}(\sigma) = \hat{x}_\sigma$ where the error $\hat{x}_\sigma - x_\sigma$ can be large when $\sigma$ is small. The extrapolation estimator applied to $\hat{f}$, which we seek to analyse, is then $\Pi_\Sigma[\hat{f}]$. For any norm $\|\cdot\|$, the total error can be decomposed as

$$\underbrace{\|f(0) - \Pi_\Sigma[\hat{f}]\|}_{\text{actual error}} \leq \underbrace{\|f(0) - \Pi_\Sigma[f]\|}_{\text{extrapolation error}} + \underbrace{\|\Pi_\Sigma[f] - \Pi_\Sigma[\hat{f}]\|}_{\text{numerical error}}. \tag{4}$$

As before we fix a reference design $\Sigma_{\text{ref}} = \{\sigma_i\}_{i=1}^{n}$ and consider $\Sigma_h = \{h\sigma_i\}_{i=1}^{n}$ for $h \in (0, 1]$. From Theorem 1 we have a bound on the extrapolation error term in (4). For the numerical error term we have the following bound, whose proof is contained in Section A.3:

---

[2]For carefully chosen $\Sigma$ (e.g. Chebyshev nodes) the Lebesgue constant $\lambda(0, \Sigma)$ grows as $O(\log(n))$, so the main concern is the term $(h\sigma_{\max})^n \lambda_{\min}(A)^{-(n+1)}$.

[3]The computational complexity is $O(n)$, but if one has access to $n$ parallel processors the computational time becomes $O(1)$.

**Theorem 2** (Numerical error bound). *In the setting of Theorem 1,*

$$\underbrace{\|\Pi_{\Sigma_h}[f] - \Pi_{\Sigma_h}[\hat{f}]\|_\infty}_{\text{numerical error}} \le \lambda(0; \Sigma_{\text{ref}}) \|f - \hat{f}\|_{\infty, \Sigma_h}.$$

The result is intuitively clear; since the extrapolation estimator is a projection and projections are linear maps, the error in the extrapolation estimator is linear in the numerical error $f - \hat{f}$ associated to the dataset. The magnitude of $\|f - \hat{f}\|_{\infty, \Sigma_h}$ is typically determined by $\|f(h\sigma_{\min}) - \hat{f}(h\sigma_{\min})\|_\infty$ where $\sigma_{\min}$ is the smallest element of $\Sigma_{\text{ref}}$. Further, for direct solvers the magnitude of the relative error is typically determined by the product of machine precision and the condition number (Weiss et al., 1986):

$$\frac{\|f(h\sigma_{\min}) - \hat{f}(h\sigma_{\min})\|_\infty}{\|f(h\sigma_{\min})\|_\infty} \lesssim \epsilon_{\text{MP}} \kappa(A + h\sigma_{\min}I) \tag{5}$$

where $\epsilon_{\text{MP}}$ is the machine precision (e.g. typically double precision; $10^{-16}$). Combining these observations, we obtain

$$\underbrace{\|\Pi_{\Sigma_h}[f] - \Pi_{\Sigma_h}[\hat{f}]\|_\infty}_{\text{numerical error}} \lesssim \lambda(0; \Sigma_{\text{ref}}) \epsilon_{\text{MP}} \kappa(A + h\sigma_{\min}I) \|f(h\sigma_{\min})\|_\infty \tag{6}$$

and to arrive at a practical upper-bound we use

$$\|f(h\sigma_{\min})\|_\infty \approx \|f(0)\|_\infty \lesssim \lambda_{\min}(A)^{-1} \|b\|_2 \tag{7}$$

Combining Theorems 1 and 2 and Equations (6) and (7) as in (4) we obtain a practically-relevant overall error bound which can be used to select $h$, as explained next.

## 2.4 Selecting $h$

The aim now is to select $h$ for which the bound obtained from (4) and Theorems 1 and 2 and their subsequent discussion is minimised. Specifically, we seek to balance the size of the the numerical error and the extrapolation error, i.e.

$$(1 + \lambda(0; \Sigma_{\text{ref}})) \sigma_{\max}^n \lambda_{\min}(A)^{-(n+1)} \|b\|_2 h^n = \lambda(0; \Sigma_{\text{ref}}) \epsilon_{\text{MP}} \kappa(A + h\sigma_{\min}I) \lambda_{\min}(A)^{-1} \|b\|_2$$

and to this end we propose to use the value of $h$ for which

$$\frac{\kappa(A + h\sigma_{\min}I)}{h^n} = \frac{(1 + \lambda(0; \Sigma_{\text{ref}}))}{\lambda(0; \Sigma_{\text{ref}})} \frac{\sigma_{\max}^n}{\lambda_{\min}(A)^n \epsilon_{\text{MP}}} \tag{8}$$

The left hand side of (8) is monotonic in $h$, diverging as $h \to 0$ and vanishing as $h \to \infty$, and thus there exists a unique solution $h_\star$ to (8). For well-conditioned $A$, (8) implies that

$$h_\star \propto (\epsilon_{\text{MP}} \kappa(A))^{1/n} \frac{\lambda_{\min}(A)}{\sigma_{\max}} \asymp \epsilon_{\text{MP}}^{1/n}, \tag{9}$$

where we have treated the Lebesgue functions as constants in the proportionality statement. Conversely, for near-singular $A$ we have $\kappa(A + h\sigma_{\min}I) \asymp \lambda_{\max}(A)/(h\sigma_{\min})$ and thus an analogous calculation based on (8) implies that

$$h_\star \asymp \epsilon_{\text{MP}}^{1/(n+1)}. \tag{10}$$

As expected, (10) is larger than (9) in general, indicating that for near-singular $A$ a larger nugget is required. More interestingly, we see that extrapolation with a larger number $n$ of data enables larger nuggets to be used; i.e. extrapolation can confer additional numerical stability, as well as accuracy, by enabling the size of the nuggets to be increased. Typical instances of the minimum critical size of nugget $\sigma_\star = h_\star \sigma_{\min}$ are depicted in Figure 2.

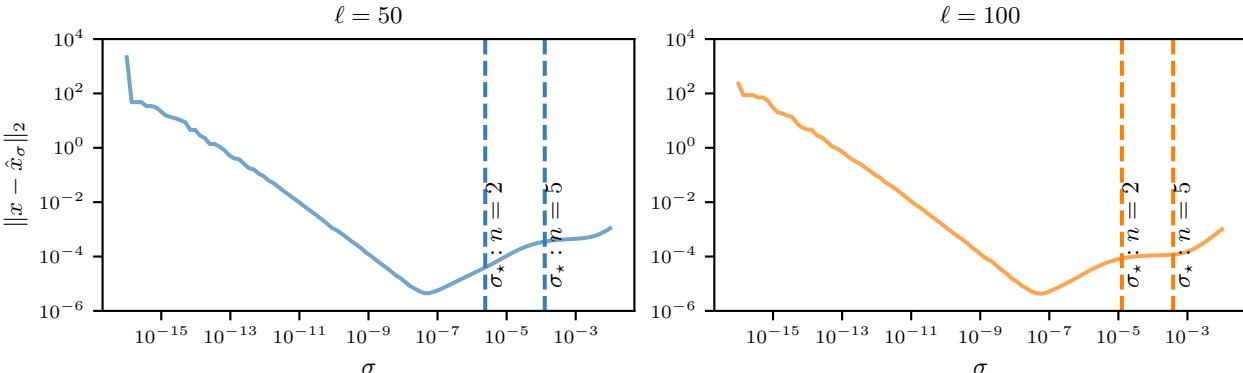

Figure 2: Identification of the critical value $\sigma_\star$, below which $\hat{x}_\sigma$ fails to be a reliable approximation to $x_\sigma$, using the approach proposed in Section 2.4. Here the matrix $A$ was a Gram matrix associated to a kernel with length-scale $\ell$; larger values of $\ell$ are associated with $A$ being more ill-conditioned. The numerical approximations were obtained using `linalg.solve` in `numpy`, a direct method based on LU factorisation. Our estimator $\sigma_\star$ is subject to slight fluctuations owing to the stochastic minimum eigenvalue approximation, but it always appears to be slightly risk-averse, favouring slightly larger nuggets than are strictly needed, which we view as a desirable property of the method.

Calculating the minimum eigenvalue of $A$ is typically as hard as solving the linear system itself, but for a symmetric positive definite matrix $M$ the smallest eigenvalue can be approximated based on the identity

$$\lambda_{\min}(M)^{-1} = \frac{1}{\inf_{\|v\|_2=1} \|Mv\|_2} = \sup_{\|v\|_2=1} \|Mv\|_2^{-1}$$

and substituting the supremum for a maximum over a finite set. For `autonugget` we used $\min\{100, 0.1d\}$ vectors $v$ randomly sampled from standard normal distribution, though more sophisticated methods are available in certain settings (Drmač, 2006; Ye, 2018).

## 3 The `autonugget` Package

A `Python` implementation of `autonugget` can be installed from `Github`[4]. For standard usage, suitable for rapid prototyping, one simply replaces the conventional direct method, e.g.

$$x = \texttt{np.linalg.solve(A,b)} \tag{11}$$

with

$$x = \texttt{autonugget(A,b)}$$

where default settings of `autonugget` are used. The reference design $\Sigma_{\text{ref}}$ can optionally be specified, as described in Section B. For simplicity we set $\Sigma_{\text{ref}} = \{2^j \sigma_{\min} : 0 \le j \le m\}$ where $m$ is user-specified (defaults to $m = 1$), since such geometric grids are known to be well-suited to polynomial extrapolation (Liem and Shih, 1995).

A distinguishing feature of `autonugget` is that it is `JAX`-compatible, enabling end-to-end training of machine learning algorithms, as will now be explained. In a slight overloading of notation, suppose that $x_\theta$ denotes the solution of the linear system defined by $A \equiv A_\theta$ and $b \equiv b_\theta$, where $A_\theta$ and $b_\theta$ depend smoothly on parameters $\theta \in \mathbb{R}^p$. This situation occurs routinely in kernel methods, for example (Rasmussen and Williams, 2006). It can then be useful to compute gradients $\nabla_\theta x_\theta$, but automatic differentiation through linear solvers can be non-trivial, for instance when logical termination criteria are used. To address this point, we employ calculus to see that

$$\nabla_\theta x_\theta = -A_\theta^{-1}(\nabla_\theta A_\theta)A_\theta^{-1}b_\theta + A_\theta^{-1}(\nabla_\theta b_\theta) \tag{12}$$

---

[4]https://github.com/hegdedisha/autonugget

which can be recursively computed with three distinct calls to `autonugget`. This calculation is implemented as a custom forward differentiation rule for `JAX` within `autonugget`. This enables automatic selection of appropriate nuggets independently for each of the three linear systems in (12) that must be numerically solved, in contrast to naïve differentiation through `autonugget` for which no such protection would be provided.

## 4 Empirical Assessment

This section presents an empirical assessment of `autonugget`, reporting results across a spectrum of linear systems from well-conditioned to ill-conditioned. Our baselines and assessment protocol are laid out in Section 4.1. The accuracy of approximations to both the solution (Section 4.2) and derivatives of the solution (Section 4.3) are examined, and the practical benefit of `autonugget` is demonstrated in Section 4.4.

### 4.1 Baselines

Since our use-case is rapid prototyping, we do *not* compare against numerical methods that are optimised for specific tasks where additional problem structure can be exploited. Rather, we compare `autonugget` with the following generic baselines (which do not employ extrapolation):

- `LU`: The standard `numpy` solver (11), which is based on LU decomposition, with no nugget.

- `LU-fix`: As `LU` but with (arbitrary) fixed nugget at single precision; $\sigma = \epsilon_{\text{sing}} = 10^{-8}$.

- `LU-cond`: As `LU` but with a nugget $\sigma$ chosen just large enough that $\kappa(A + \sigma I) < \epsilon_{\text{sing}}^{-1} = 10^8$.

- `LU-adapt`: As `LU` but with a nugget $\sigma = \sigma_\star$ as in Section 2.4.

- `CG`: The conjugate gradient method (Hestenes et al., 1952) with `scipy` implementation, with default relative tolerance of $10^{-5}$.

- `LSTSQ`: Least squares solution (i.e. pseudo-inverse), implemented using `numpy`.

- `SVD`: Solution using Singular Value Decomposition, implemented using `numpy`.

- `TSVD`: Solution using truncated SVD, where singular values are truncated such that all of them are greater than $10^{-8}$, modified from `numpy`-SVD.

`JAX` versions of `LU`, `LU-fix`, `CG`, `LSTSQ`, `SVD`, `TSVD` are used for derivate computations and are implemented on CPU.

As discussed in Section 1, Tikhonov regularisation helps address two separate concerns – regularisation and ill-conditioning. While some of the regularisation-based methods listed above (e.g., `LU-fix`, `LU-cond`, `LU-adapt`, `LSTSQ`, `TSVD`) are also helpful for the former purpose, here we strictly test their performance in solving ill-conditioned positive-definite matrices as compared to `autonugget`.

To automatically generate a set of test problems we consider Gram matrices $A$ associated to a positive definite kernel $k : \mathbb{R} \times \mathbb{R} \to \mathbb{R}$ and a (fixed) collection of uniformly spaced distinct nodes $\{z_i\}_{i=1}^d \subset \mathbb{R}$, so that $A_{i,j} = k(z_i, z_j)$ for all $i, j \in \{1, \ldots, d\}$. Specifically, we take $k(z, z') = \exp(-(z - z')^2/\ell^2)$, unless specified otherwise, since for this kernel changes in the length-scale $\ell$ strongly affect the condition of $A$ (larger $\ell$ is associated with larger $\kappa(A)$). Again, we emphasise that for any particular set of test problem one can design bespoke numerical methods (indeed, numerical methods for kernel matrices are well-studied, see Section 1.2 of Schafer et al., 2021, for a recent review); our interest is specifically in generic methods which are agnostic to any special structure present in $A$.

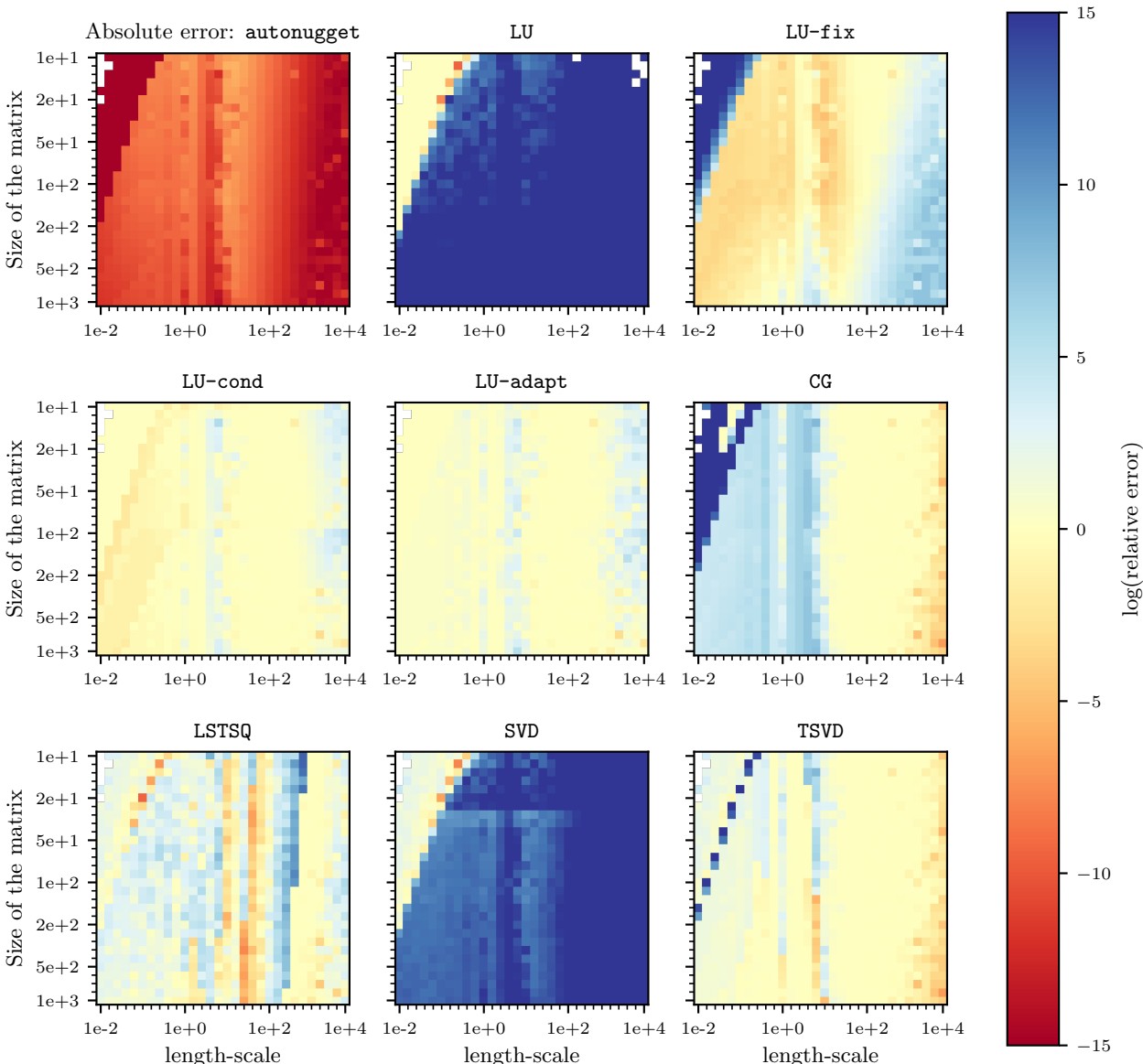

Figure 3: Solver accuracy assessment. Here we compared `autonugget` to our baselines, varying the dimension $d$ of the matrix and the length-scale $\ell$ of the kernel. The logarithm of the error relative to `autonugget` is reported; blue indicates that `autonugget` out-performed the baseline method. [Alternative versions of this figure are included in the Supplement for (a) a different choice of kernel (Fig. 7), and (b) different polynomial orders for extrapolation (Fig. 9).]

## 4.2 Solver Accuracy

To test solver accuracy we fix the solution vector $x$ to be the vector $\mathbf{1}$ whose entries are each 1, and set $b = Ax$. Given an approximate solution $\hat{x}$ produced by a numerical method, the error is quantified as $\|\hat{x} - x\|_2$.

Figure 3 shows log of relative error with respect to `autonugget`, computed as the ratio of error of the method to the error in `autonugget`. Absolute errors are presented in the appendix in Fig. 6. The positive values (in blue) indicate that `autonugget` out-performed the baseline method. It can be seen that `autonugget` performs far better than `LU` and `SVD` for most of the linear systems considered, especially the ill-conditioned

systems (i.e. large length-scale $\ell$). `autonugget` also out-performed `CG` (with the default tolerance used), `LSTSQ`, `TSVD` albeit to a lesser extent. However, we note that the absolute error of `autonugget` is still low even for the cases where it is slightly outperformed by one of the other methods. More importantly, we observe that `LU-fix` performed worse than `autonugget` on both well-conditioned systems (where the nugget is too large, introducing a bias) and ill-conditioned systems (where the nugget is too small, meaning that insufficient regularisation is provided); this clearly illustrates the need for *adaptive* selection of a nugget. However, the `LU-adapt` baseline was also out-performed by `autonugget`; this demonstrates the additional accuracy that comes from the extrapolation functionality of `autonugget`.

For the experiments we report in the main text we employed extrapolation based on $n = 2$ linear solves (i.e. linear extrapolation) as a default within `autonugget`. As ablation studies, this experiment was repeated for (a) a different choice of kernel (Fig. 7), (b) set of non-kernel type matrices (Fig. 8) and (c) different numbers of data $n$ for extrapolation (Fig. 9), with results contained in the Supplement. Improved accuracy was observed in some cases with $n > 2$, but we found the benefit was not substantial enough to merit the additional computational cost as a default. To show randomised eigenvalue approximation did not affect conclusions, we show Fig. 3 computed with a different seed in Fig. 10. A table of mean and standard deviation of mean absolute errors (across different seeds, with randomly spaced $\{z_i\}$) is listed in Table 1 and Table 2.

### 4.3 Derivative Accuracy

For the second set of experiments we sought to evaluate the accuracy with which the derivative of the solution of the linear system was computed. For this purpose we consider a parametrised linear system $A \equiv A_\theta$, $b \equiv b_\theta$, where to enable an analytically tractable gold-standard we employed a simple dependence on the parameter $\theta$ as $A = \theta \tilde{A}$ and $b = \tilde{b}$ for a fixed $\tilde{A}$ and $\tilde{b}$. As such, the true derivative is $\nabla_\theta x_\theta = -\frac{x}{\theta}$. Given a numerical approximation $\nabla \hat{x}_\theta$ to the derivative $\nabla x_\theta$, the error was quantified using $\|\nabla \hat{x}_\theta - \nabla x_\theta\|_2 / d$, where we have normalised for the dimension $d$ to make comparison more straightforward.

Figure 4 compares the accuracy of derivatives obtained from `autonugget` to automatic differentiation through other baselines using `JAX`. It can be seen that `LU` performed worst; we attribute this to the absence of any appropriate regularisation during the three linear system solves that are required when automatic differentiation is performed. `LU-cond` and `LU-adapt` performed much better than `LU`, but were still out-performed by `autonugget` when the length-scale is large and the linear-system is severely ill-conditioned. `autonugget` also performed better than the other baselines, as similar to the solvers in the previous section, with the difference being more pronounced due to multiple linear solves involved in the derivative calculation. Interestingly, `LSTSQ`, `SVD` and `TSVD` failed to reach finite values for ill-conditioned systems, rendering them impractical for multiple systems. This highlights the advantage of using our custom auto differentiation provided by `autonugget` that enables efficient derivate computation involving multiple separate `autonugget` calls.

### 4.4 Application to GP Regression

Finally, we test `autonugget` in a quasi-realistic setting where rapid prototyping might be required; a hyperparameter optimisation problem for a Gaussian process (GP) regression task. As both a model and a data-generating process we took a centred GP with the squared exponential kernel and generated (noiseless) data on a regular grid of $5 \times 5$ points in $[0, 1]^2$. For data generation we used a kernel with length-scale $\ell = 50$. The task is to infer a suitable length-scale $\ell$ from the dataset by using leave one out cross-validation based on the predictive root-mean squared error. Since the true length-scale is much larger than the domain on which data were generated, such optimisation requires working with kernel matrices which, despite being small, are severely ill-conditioned (Lin et al., 2024). For optimisation we consider a gradient-descent method with adaptive step-sizes computed from backtracking line-search (Nocedal and Wright, 2006). Taking the gradient of the cross-validation loss requires differentiating the solution of a linear system involving the kernel matrix; as in Section 4.3 we compare `autonugget` to automatic differentiation through either `LU` or `LU-fix` using `JAX`.

Figure 5 shows a clear advantage to using `autonugget` in this setting. Indeed, the loss function computed using `autonugget` has a clear minimum around the true length-scale $\ell = 50$, and the accurate gradient

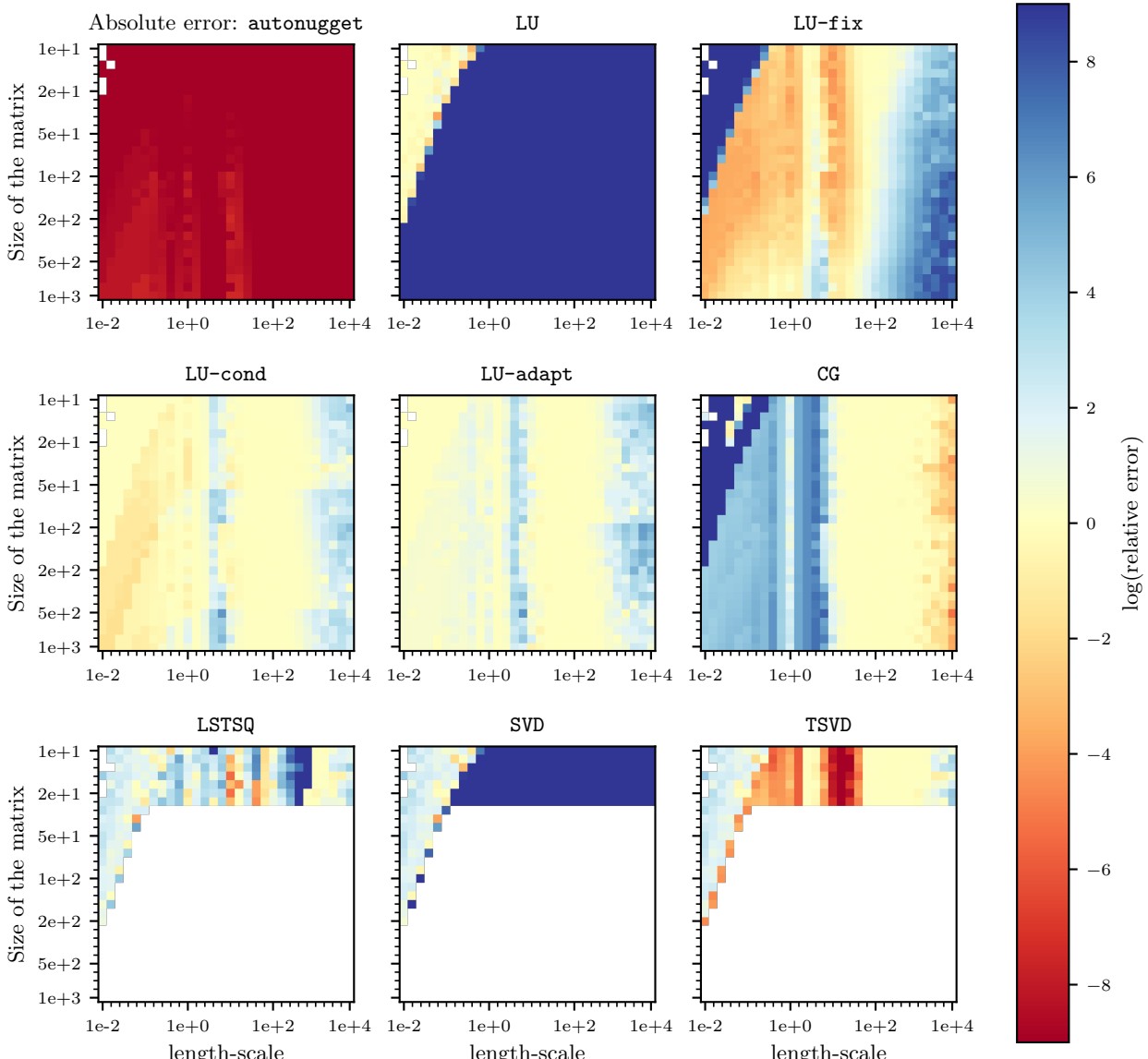

Figure 4: Derivative accuracy assessment. Here we compare `autonugget` to the derivative computed using `JAX` applied to the baselines `LU` and `LU-fix`, varying the dimension $d$ of the matrix and the length-scale $\ell$ of the kernel. The logarithm of the (dimension normalised) error relative to `autonugget` is reported; blue indicates that `autonugget` out-performed the baseline method.

calculation ensures rapid convergence of the optimisation method. In contrast, the loss functions computed using `LU` and `SVD` are non-smooth (due to the lack of any Tikhonov regularisation) and the optimiser diverges to a pathologically large value of $\ell$. `LSTSQ` and `TSVD` also lead non-smooth loss functions, and the optimiser barely moves and converges to the initial point. `LU-fix` (i.e. the use of a fixed-size nugget across all length-scales, a common approach in rapid prototyping) confers stable optimisation, but the optimiser converges to a value for the length-scale that is far too small. This occurs due to the bias introduced by the nugget, which encourages the data to be (incorrectly) explained as "noise" instead of signal. While `LU-cond` and `LU-adapt` managed to converge to appropriate length-scales in this experiment, it can be seen that their computed cross-validation loss increases as the length-scale is increased; this is incorrect, because the cross validation loss should become constant as $\ell \to \infty$. This increase is an artefact of an increasing amount of regularisation being applied in `LU-cond` and `LU-adapt` as the increasing length-scale $\ell$ increases the condition number of

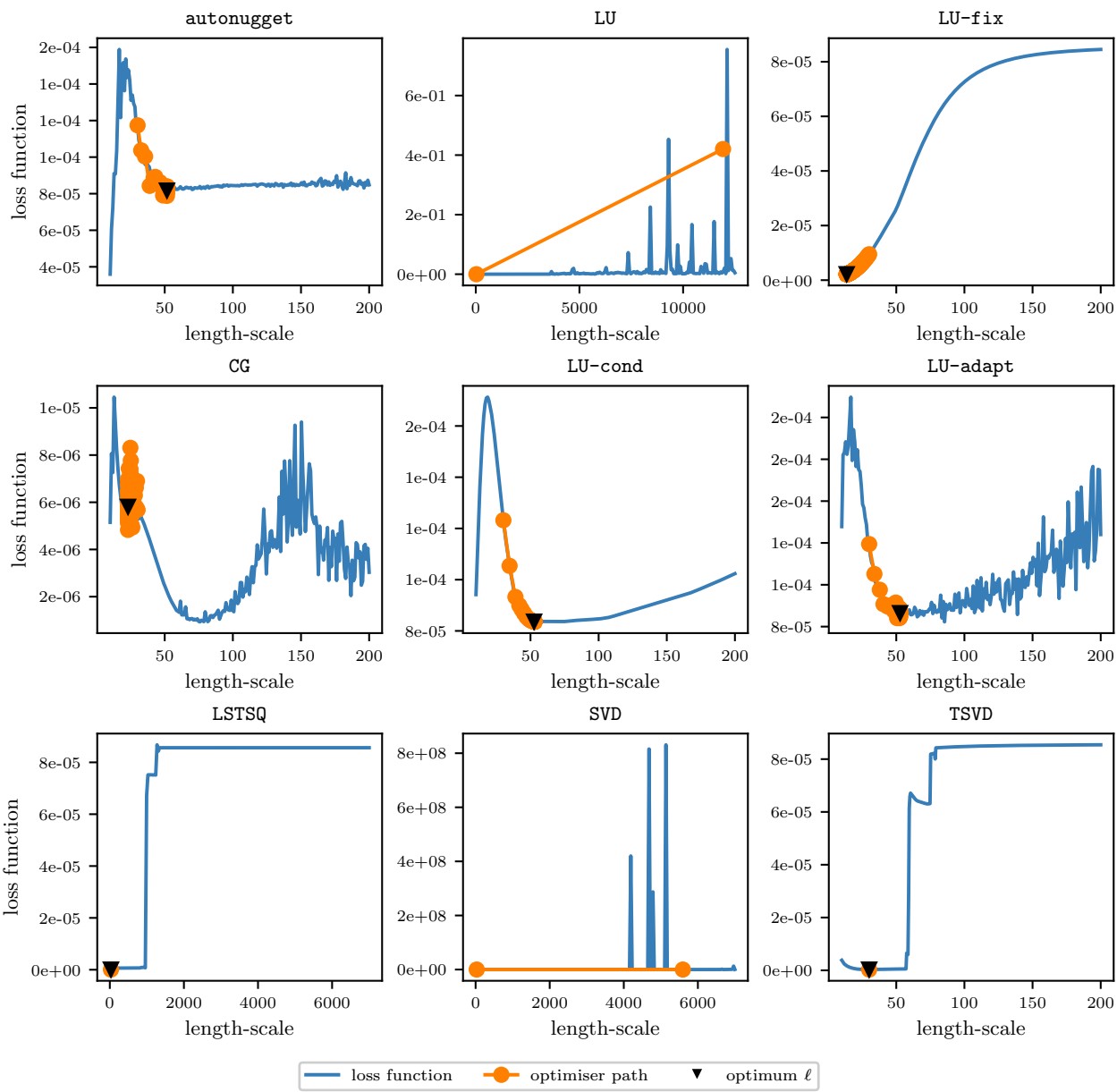

Figure 5: Application to a Gaussian process regression task. A gradient-based optimiser was applied to the cross-validation loss function described in Section 4.4, with evaluations of the loss function and its gradients performed using either `autonugget` or one of the baselines described in Section 4.1.

the associated linear systems. In contrast, the constant tail behaviour of the cross-validation loss is correctly captured by `autonugget`.

## 5 Discussion

Numerical linear algebra is critical to accelerating computations in machine learning, for instance in kernel methods (Rudi et al., 2017), GPs (Chen et al., 2025), and deep learning (Sato et al., 2024). Yet sophisticated numerical methods are not the first port-of-call during methodological development. Motivated by the lack of a tuning-free, automatic differentiation-compatible routine to solve ill-conditioned linear systems of equations, we introduced `autonugget`. A distinguishing feature of `autonugget` is its use of extrapolation

to gain accuracy beyond the critical value of $\sigma$ at which explosive behaviour is encountered in solution using a direct method. An empirical assessment supported the use of `autonugget` as a general-purpose tool for solving linear systems of equations specified by symmetric positive definite matrices, which are widely encountered in machine learning.

`autonugget` is currently limited to symmetric positive definite matrices, and a future direction would be extend our theoretical framework to non-SPD matrices. A particularly interesting case would be the behaviour of extrapolated solution as $\sigma \to 0$ in the cases where a unique solution does not exist, e.g., when $A$ is symmetric positive semidefinite. While `autonugget` performs better than standard methods for ill-conditioned methods, it also incurs higher computational cost. A more efficient low-level implementation of `autonugget` can potentially address this concern. Additionally, as a low-cost alternative, we plan explore an SVD-centric approach that would use a single SVD computation with extrapolation in the span of SVD-basis, and the corresponding error analysis. The use cases of `autonugget` are limited to matrices that can be stored in memory, but for prototyping applications this is often sufficient. Another possible direction for future work is to incorporate probabilistic error bars for the extrapolated solution, for instance by replacing polynomial extrapolation with a statistical regression model (Oates et al., 2025).

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

## Appendices

Section A contains proofs for all theoretical results appearing in the main text. Section B contains documentation for `autonugget`. Section C reports additional experimental results, along with full details for the experiment that were performed.

## A  Proofs

Section A.1 contains preliminary results which will be required to prove Theorem 1 and Theorem 2. Theorem 1 is proven in Section A.2, while Theorem 2 is proven in Section A.3.

### A.1  Preliminary Results

For convenience we introduce the shorthand $\|f\|_{\infty,\sigma_{\max}} := \|f\|_{\infty,[0,\sigma_{\max}]} = \sup_{\sigma \in [0,\sigma_{\max}]} |f(\sigma)|$ on $C^0([0,\sigma_{\max}],\mathbb{R})$. Further, following the same notation used in Section 2.3 of the main text, let

$$\Pi_\Sigma : C^0([0,\sigma_{\max}],\mathbb{R}) \to \mathbb{R}$$
$$f \mapsto f_n(\mathbf{0})$$

where $f_n$ denotes the interpolating polynomial (in $\mathcal{P}$) of $f$ on $\Sigma$.

**Proposition 1** (Polynomial extrapolation error). *Let $f \in C^0([0,\sigma_{\max}],\mathbb{R})$. Let $\Sigma$ be $\mathcal{P}$-unisolvent and let $f_n \in \mathcal{P}^d$ interpolate $f$ on $\Sigma$. Then*

$$|f(0) - f_n(0)| \leq (1 + \lambda(0;\Sigma)) \inf_{p \in \mathcal{P}} \|f - p\|_{\infty,\sigma_{\max}}.$$

*Proof of Proposition 1.* Let $p_\star$ attain the infimum. Then by the triangle inequality, and the fact that $\Pi_\Sigma$ is a projection,

$$\begin{aligned}
|f(0) - f_n(0)| &\leq |f(0) - p_\star(0)| + |p_\star(0) - f_n(0)| \\
&= |f(0) - p_\star(0)| + |\Pi_\Sigma(p_\star) - \Pi_\Sigma(f)| \\
&= |f(0) - p_\star(0)| + |\Pi_\Sigma(p_\star - f)| \\
&\leq \|p_\star - f\|_{\infty,\sigma_{\max}} + \|\Pi_\Sigma\|_{\mathrm{op}}\|p_\star - f\|_{\infty,\sigma_{\max}} = (1 + \|\Pi_\Sigma\|_{\mathrm{op}})\|p_\star - f\|_{\infty,\sigma_{\max}}
\end{aligned}$$

and the result follows from

$$\|\Pi_\Sigma\|_{\mathrm{op}} = \sup_{g \neq 0} \frac{|\Pi_\Sigma(g)|}{\|g\|_{\infty,\sigma_{\max}}} = \sup_{g \neq 0} \frac{|\sum_{i=1}^n g(\sigma_i)\ell_i(0;\Sigma)|}{\|g\|_{\infty,\sigma_{\max}}} \tag{13}$$

since the right hand side of (13) is at most $\lambda(0;\Sigma)$. $\qquad\square$

The following result establishes convergence acceleration for Richardson extrapolation:

**Proposition 2** (Convergence acceleration in general). *Let $f \in C^0([0,\sigma_{\max}],\mathbb{R})$ and assume that there exist coefficients $\beta_i$ for which the residual*

$$R(\sigma) = f(\sigma) - \sum_{i=1}^m \beta_i p_i(\sigma) \tag{14}$$

*vanishes as $\sigma \to 0$. Let $\Sigma_{\mathrm{ref}}$ be $\mathcal{P}$-unisolvent. Let $\Sigma_h = \{h\sigma_i\}_{i=1}^n$ for $h \in (0,1]$. Let $f_n^h \in \mathcal{P}^d$ interpolate $f$ on $\Sigma_h$. Then*

$$\underbrace{|f_n^h(0) - f(0)|}_{\text{extrapolation error}} \leq \underbrace{(1 + \lambda(0;\Sigma_{\mathrm{ref}}))}_{\text{constant in } h} \|R\|_{\infty,h\sigma_{\max}}.$$

In other words, if $f$ admits a Taylor expansion at $\sigma = 0$ and $\mathcal{P}$ contains more than just the leading constant term in this Taylor expansion, then $f_n^h(0)$ converges faster than the original $f(\sigma)$, for any $\sigma \in \Sigma_h$, as $h \to 0$.

*Proof of Proposition 2.* Note that $f_n^h$ exists and is unique by the assumption that $\Sigma_{\text{ref}}$ is $\mathcal{P}$-unisolvent, which implies also that $\Sigma_h$ is $\mathcal{P}$-unisolvent.

From Proposition 1,

$$|f(0) - f_n^h(0)| \leq (1 + \lambda(0; \Sigma_h)) \inf_{p \in \mathcal{P}} \|f - p\|_{\infty, h\sigma_{\max}}.$$

Since $\lambda(0; \Sigma_h) = \lambda(0; \Sigma_{\text{ref}})$ for all $h > 0$, we have that

$$|f(0) - f_n^h(0)| \leq (1 + \lambda(0; \Sigma_{\text{ref}})) \inf_{p \in \mathcal{P}} \|f - p\|_{\infty, h\sigma_{\max}}.$$

The first term is constant in $h$. For the second term,

$$\inf_{p \in \mathcal{P}} \|f - p\|_{\infty, h\sigma_{\max}} \leq \left\| f - \sum_{i=1}^m \beta_i p_i \right\|_{\infty, h\sigma_{\max}} = \|R\|_{\infty, h\sigma_{\max}},$$

completing the argument. $\qquad\square$

## A.2   Proof of Theorem 1

*Proof of Theorem 1.* Let $f_i$ denote the $i$th coordinate of $f$. Since $f_i \in C^\infty([0, \sigma_{\max}], \mathbb{R})$, from the mean value theorem the Taylor expansion of $f_i$ at $\sigma = 0$ with residual

$$R_i(\sigma) = f_i(\sigma) - \sum_{i=1}^n \beta_i \sigma^{i-1}$$

satisfies

$$|R_i(\sigma)| \leq \frac{\sigma^n}{n!} \sup_{\tilde{\sigma} \in [0, \sigma]} |f_i^{(n)}(\tilde{\sigma})|.$$

From calculus

$$f_i^{(n)}(\sigma) = (-1)^n (n!)[(A + \sigma I)^{-(n+1)} b]_i,$$

so, using a spectral bound,

$$|R_i(\sigma)| \leq \sigma^n \sup_{\tilde{\sigma} \in [0, \sigma]} |[(A + \tilde{\sigma} I)^{-(n+1)} b]_i|$$

$$\leq \sigma^n \sup_{\tilde{\sigma} \in [0, \sigma]} \lambda_{\min}(A + \tilde{\sigma} I)^{-(n+1)} \|b\|_2$$

$$\implies \|R(\sigma)\|_\infty \leq \sigma^n \sup_{\tilde{\sigma} \in [0, \sigma]} \lambda_{\min}(A + \tilde{\sigma} I)^{-(n+1)} \|b\|_2.$$

Since $A \prec A + \sigma I$, we have $\lambda_{\min}(A) \leq \lambda_{\min}(A + \sigma I)$, and

$$\|R(\sigma)\|_\infty \leq \sigma^n \lambda_{\min}(A)^{-(n+1)} \|b\|_2.$$

Finally,

$$\sup_{\sigma \in [0, h\sigma_{\max}]} \|R(\sigma)\|_\infty \leq (h\sigma_{\max})^n \lambda_{\min}(A)^{-(n+1)} \|b\|_2.$$

From Proposition 2 we have the result. $\qquad\square$

### A.3 Proof of Theorem 2

*Proof of Theorem 2.* From the definition of the operator norm in (2),

$$\|\Pi_{\Sigma_h}[g] - \Pi_{\Sigma_h}[\hat{g}]\|_\infty \leq \|\Pi_{\Sigma_h}\|_{\mathrm{op}} \|g - \hat{g}\|_\infty \tag{15}$$

for all $g, \hat{g} \in C^0([0, \sigma_{\max}], \mathbb{R})$. Since $\Pi_{\Sigma_h}[f]$ and $\Pi_{\Sigma_h}[\hat{f}]$ depend on $f$ and $\hat{f}$ only at the inputs $\Sigma_h$, we can take an infimum on both sides of (15) over all continuous $g$ (resp. $\hat{g}$) that agree with $f(\sigma)$ (resp. $\hat{f}(\sigma)$) for all $\sigma \in \Sigma_h$, to obtain

$$\|\Pi_{\Sigma_h}[f] - \Pi_{\Sigma_h}[\hat{f}]\|_\infty \leq \|\Pi_{\Sigma_h}\|_{\mathrm{op}} \|f - \hat{f}\|_{\infty, \Sigma_h}.$$

Recall that $\sigma_{\max} := \max\{\sigma_i\}_{i=1}^n$. Using the shorthand introduced in Section A.1, the result follows from the same argument used to establish (13):

$$\|\Pi_{\Sigma_h}\|_{\mathrm{op}} := \sup_{g \neq 0} \frac{|\Pi_{\Sigma_h}(g)|}{\|g\|_{\infty, \sigma_{\max}}} = \sup_{g \neq 0} \frac{|\sum_{i=1}^n g(\sigma_i) \ell_i(0; \Sigma_h)|}{\|g\|_{\infty, \sigma_{\max}}}$$

and the fact that the right hand side of (13) is at most $\lambda(0; \Sigma_h)$ which in turn is equal to $\lambda(0; \Sigma_{\mathrm{ref}})$. $\qquad\square$

## B  Documentation for `autonugget`

The `autonugget` package can be installed from Github[5]. This mainly implements the function `autonugget` as described below:

```
autonugget(A, b, m=1, mode = 'adapt', extrap = True, JAX_enabled = False, sigma_star =
None, Sigma_ref = None)
```

This function computes the solution of the system $Ax = b$ using `autonugget`. This function is compatible with differentiation using `JAX`.

**Parameters**

`A`: array like, shape $(d, d)$
      $d \times d$ matrix
`b`: array like, shape $\{(d,), (d, k)\}$
      $d$ dimensional vector
`m`: int; optional
      desired degree of the polynomial
`mode`: str; default: `'adapt'`
      method to choose the nugget; should be one of `'adapt'` or `'cond'`
`extrap`: bool; default: `True`
      use extrapolation to calculate the approximate solution
`JAX_enabled`: bool; default: `False`
      switch to `True` to use custom forward differentiation from `JAX`
`sigma_star`: float; optional
      value of the nugget
`Sigma_ref`: list; optional
      reference design to choose `sigma_star`

**Returns**

`x`: array like, shape $\{(d,), (d, k)\}$
      approximate solution to the system $Ax = b$. Returned shape is same as shape of $b$.

Table 1: Mean and standard deviation of absolute error across multiple runs (with different seeds and randomly spaced $\{z_i\}$) averaged over the 100 ill-conditioned matrices in the bottom-right quarter for the experiment in Section 4.2. Apart from `TSVD`, various versions of `autonugget`, with and without extrapolation have the best performance compared to other baselines. `autonugget-cond` (as described in Section C.2) has the least mean absolute error after `TSVD`, followed by `autonugget`.

|                   | Mean       | Standard Deviation |
| ----------------- | ---------- | ------------------ |
| autonugget        | 1.80e-04   | 1.71e-05           |
| autonugget-cond   | **1.41e-04** | 2.91e-07         |
| LU                | 2.52e+06   | 3.72e+06           |
| LU-fixed          | 5.73e-04   | 2.90e-06           |
| LU-adapt          | 2.31e-04   | 1.61e-05           |
| LU-cond           | 1.95e-04   | 1.23e-06           |
| CG                | 4.84e-04   | 9.87e-06           |
| LSTSQ             | 5.00e-04   | 9.57e-05           |
| SVD               | 2.56e+02   | 5.60e+00           |
| TSVD              | **7.12e-06** | 6.37e-07         |

Table 2: Mean and standard deviation of error across multiple runs averaged over all 900 matrices for the experiment in Section 4.2. For this larger set of matrices, extrapolated versions of `autonugget` – `autonugget` and `autonugget-cond` perform well after `TSVD`.

|                   | Mean       | Standard Deviation |
| ----------------- | ---------- | ------------------ |
| autonugget        | 2.41e-04   | 7.68e-06           |
| autonugget-cond   | **1.20e-04** | 9.98e-07         |
| LU                | 5.50e+15   | 1.86e+16           |
| LU-fixed          | 2.27e-04   | 9.18e-07           |
| LU-adapt          | 3.89e-04   | 1.34e-05           |
| LU-cond           | 1.82e-04   | 1.18e-06           |
| CG                | 2.30e-02   | 4.73e-04           |
| LSTSQ             | 2.02e-03   | 2.31e-04           |
| SVD               | 3.79e+02   | 2.11e+02           |
| TSVD              | **9.37e-06** | 2.05e-07         |

## C  Additional Experimental Results

### C.1  Additional results from Section 4

### C.2  `autonugget-cond`

Along with the strategy explained in Section 2.4, the `autonugget` package provides an alternative strategy to choose $\sigma_\star$ - by choosing the smallest $\sigma$ such that the condition number of $(A + \sigma I)$ does not exceed $10^8$ — as a computationally cheaper alternative which only involves the calculation of condition numbers and avoids the stochastic minimum eigenvalue computations. Figs. 11 and 12 reproduce the results of Figs. 3 to 5, 7 and 9 using `autonugget-cond` in place of `autonugget`.

Fig. 11 suggests that `autonugget-cond` performs slightly better than `autonugget` for better conditioned linear systems, but `autonugget` outperforms `autonugget-cond` in highly ill-conditioned matrices, leading us to favour `autonugget`. Fig. 12 shows `autonugget-cond` results in a smoother loss function, but both the methods converge to very close optimum $\ell$.

---

[5]`https://github.com/hegdedisha/autonugget`

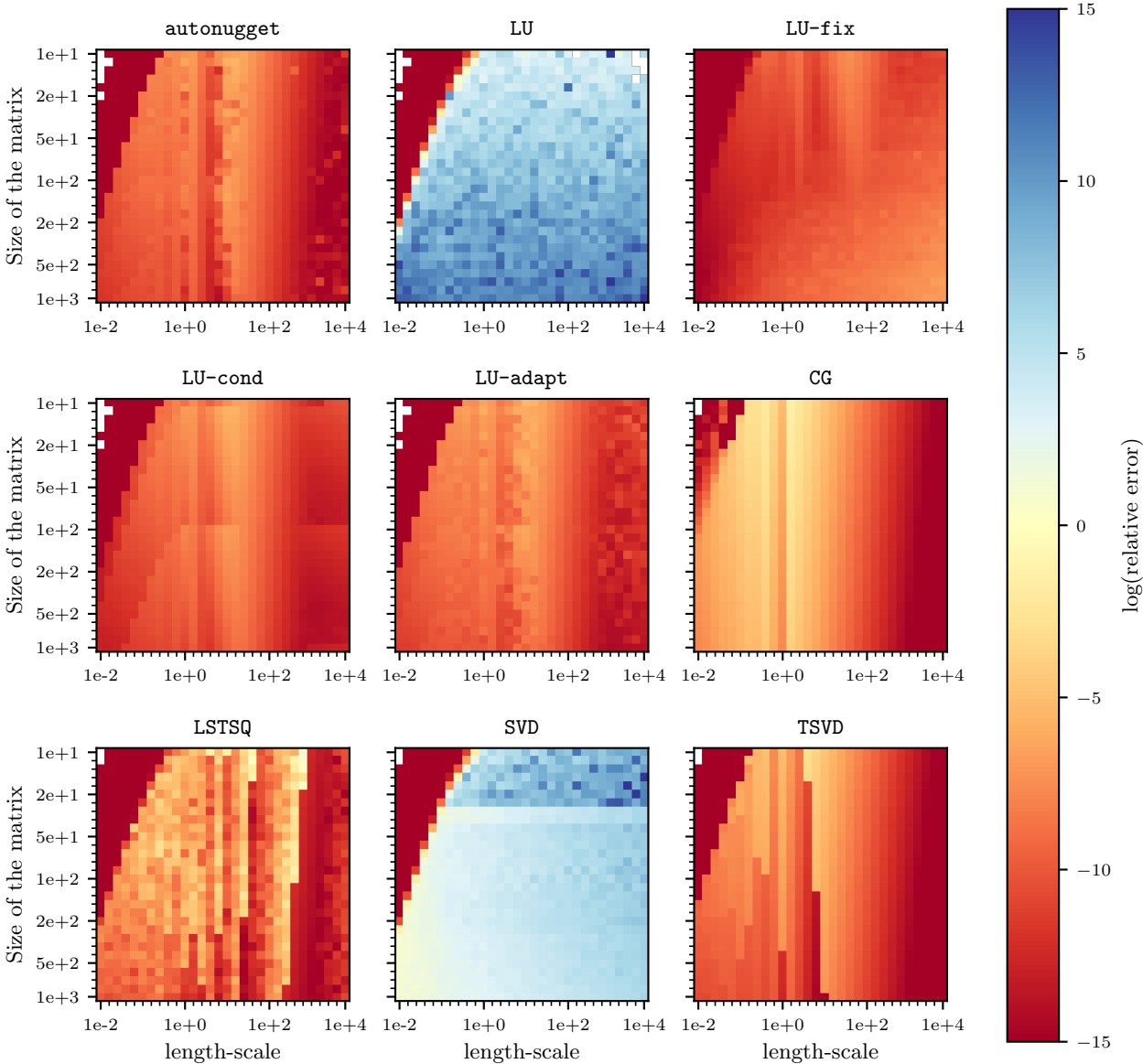

Figure 6: Absolute errors of all methods for the experiment in Fig. 3.

## C.3 Wall Times

Fig. 13 shows the total time taken by each of the methods in Section 4.2, as a function of the size of the matrix. While `autonugget` has the same scaling as solving the linear system (i.e., it scales cubically with dimension), in terms of FLOPS, it involves solving additional linear systems (for the extrapolation) and performing additional SVDs (to identify the nugget). Therefore, as we see in Fig. 13, `autonugget` does have a higher cost, but it is higher by a constant factor. For our recommended settings (extrapolation with a linear polynomial), only one additional linear solve is required, and so the additional cost is dominated by the search procedure used to identify a nugget.

To mitigate the cost of `autonugget`, when the dimension of the matrix is large we suggest the use of lower-cost alternative `autonugget-cond`, which also provides a comparable performance to `autonugget` (Fig. 11). We re-emphasise that our method has not been fully optimised for speed, so there are likely significant gains to be made here. Furthermore, the goal of our method is to provide a solver that is (possibly slow but)

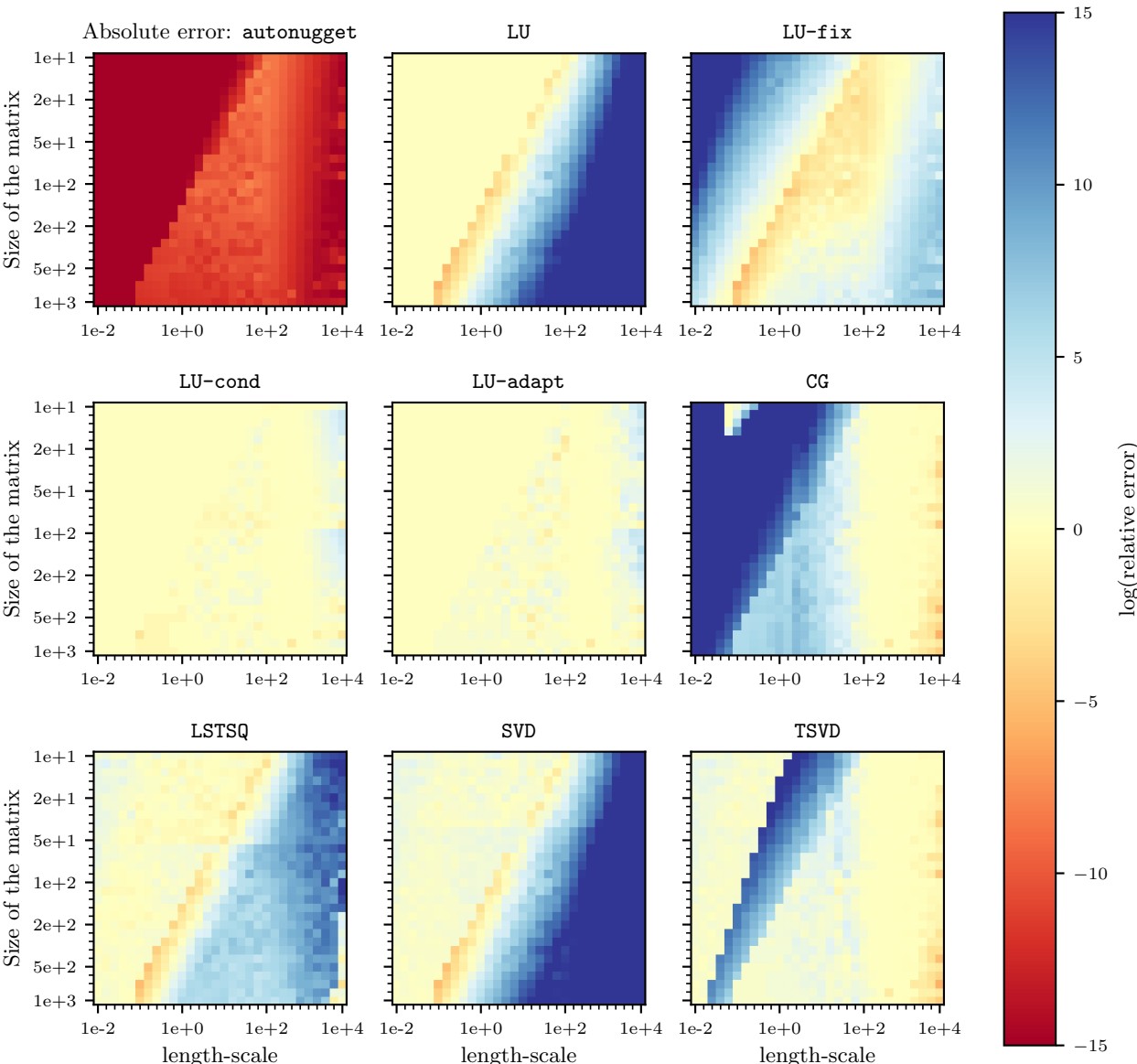

Figure 7: Results on the Matérn kernel matrices. Here we compare `autonugget` to our baselines, varying the dimension $d$ of the matrix and the length-scale $\ell$ of the kernel.

reliable, tuning-free and compatible with JAX, for prototyping and thus the higher cost of our method is a secondary concern.

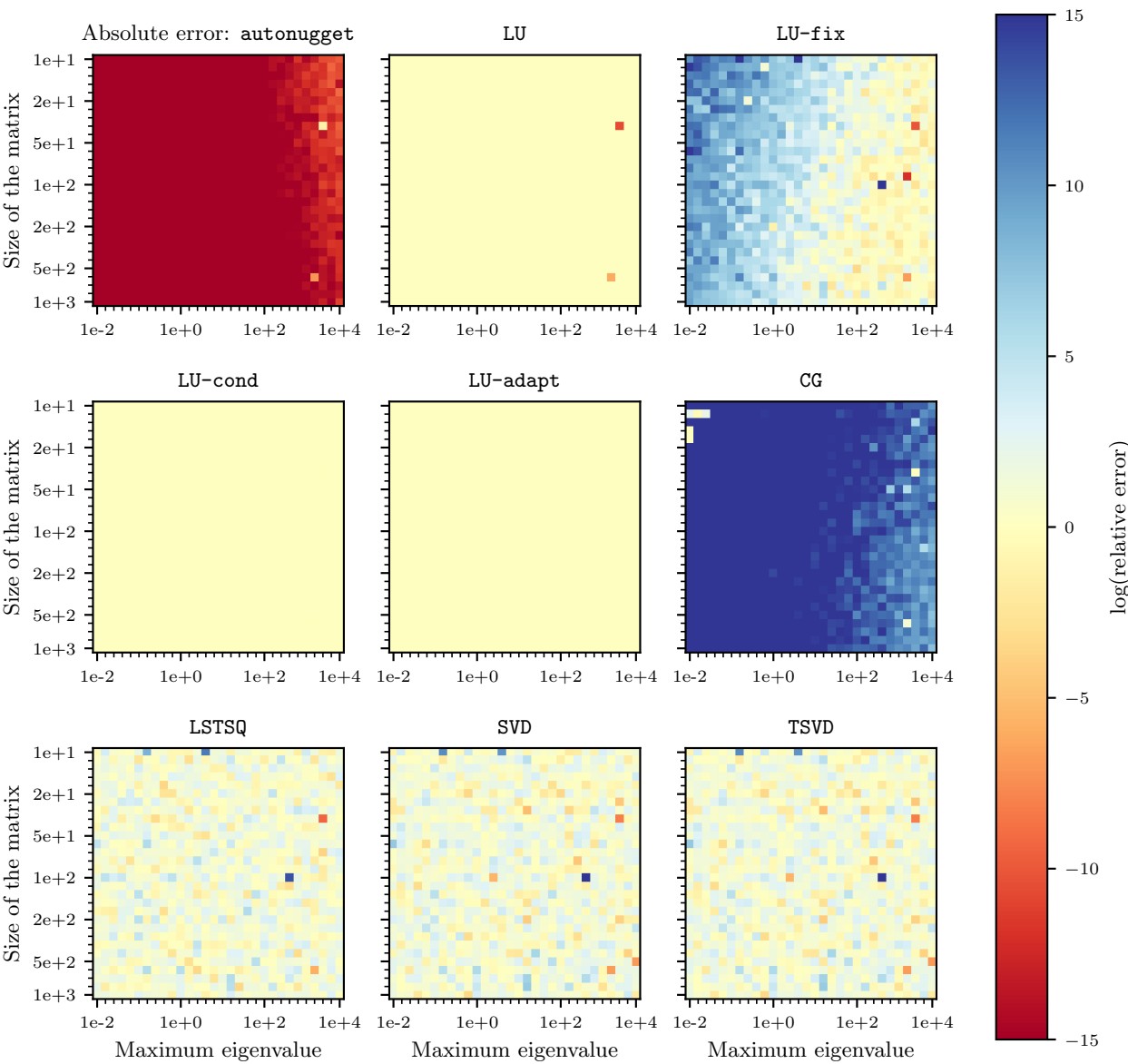

Figure 8: Results on non-kernel type matrices. Here we compared `autonugget` to our baselines, for a set of positive definite matrices generated as a product of $UDU^\top$, where $U$ is an orthogonal matrix drawn from a Haar measure, and $D$ is a diagonal matrix of eigenvalues drawn randomly from $\mathcal{U}(0.01, \text{maximum eigenvalues})$. This ensures that the matrices are ill-conditioned as the maximum eigenvalue increases. We can see that `autonugget` performs almost the same as `LU`, due to the matrices still being well-conditioned, even for an condition number of $\approx 10^{10}$. `autonugget` performs better than `LU-fix` for better conditioned matrices, due to the fixed nugget being too large. `autonugget` performs much better than `CG` as well, possibly due to the default tolerance being too high for such matrices.

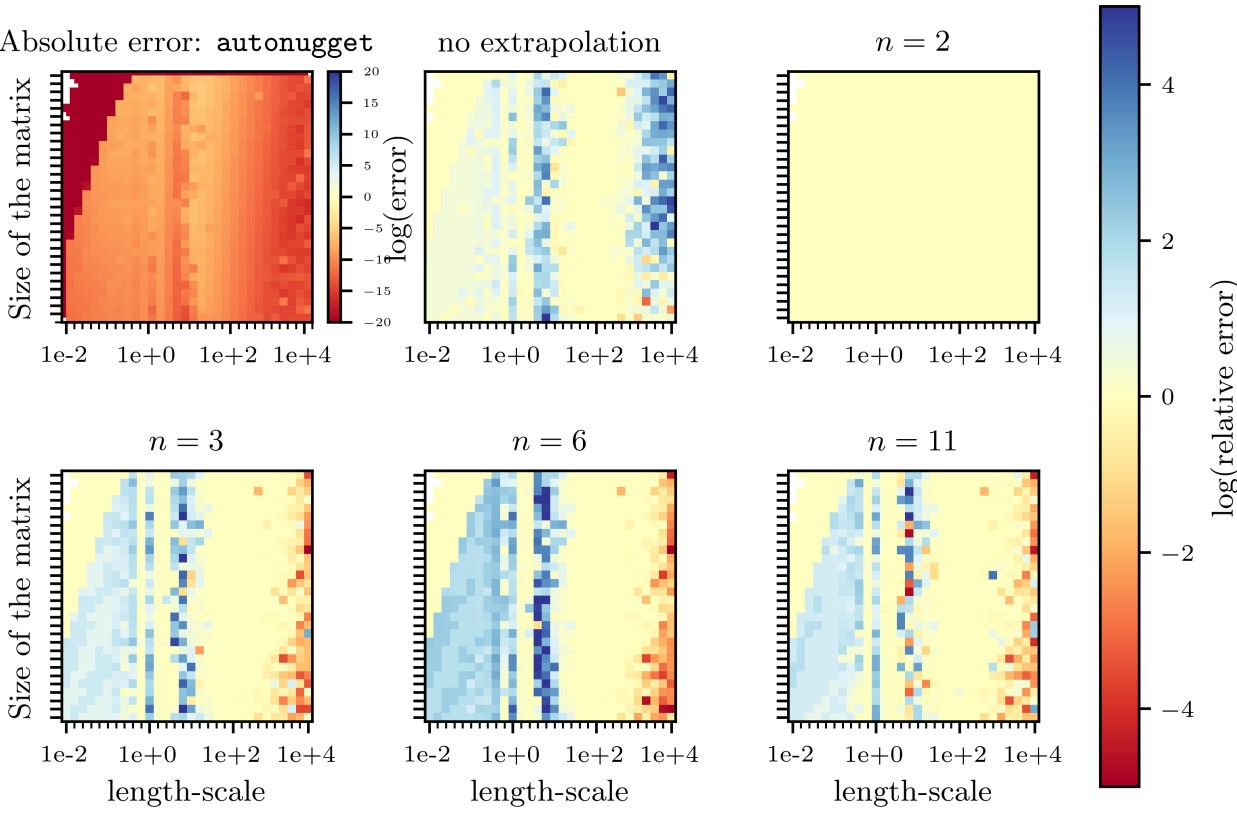

Figure 9: Results on the squared exponential kernel matrices, comparing solutions with no extrapolation and extrapolation using different degree polynomials. All the relative errors are with respect to `autonugget` with the default choice of extrapolation using 2 linear solves.

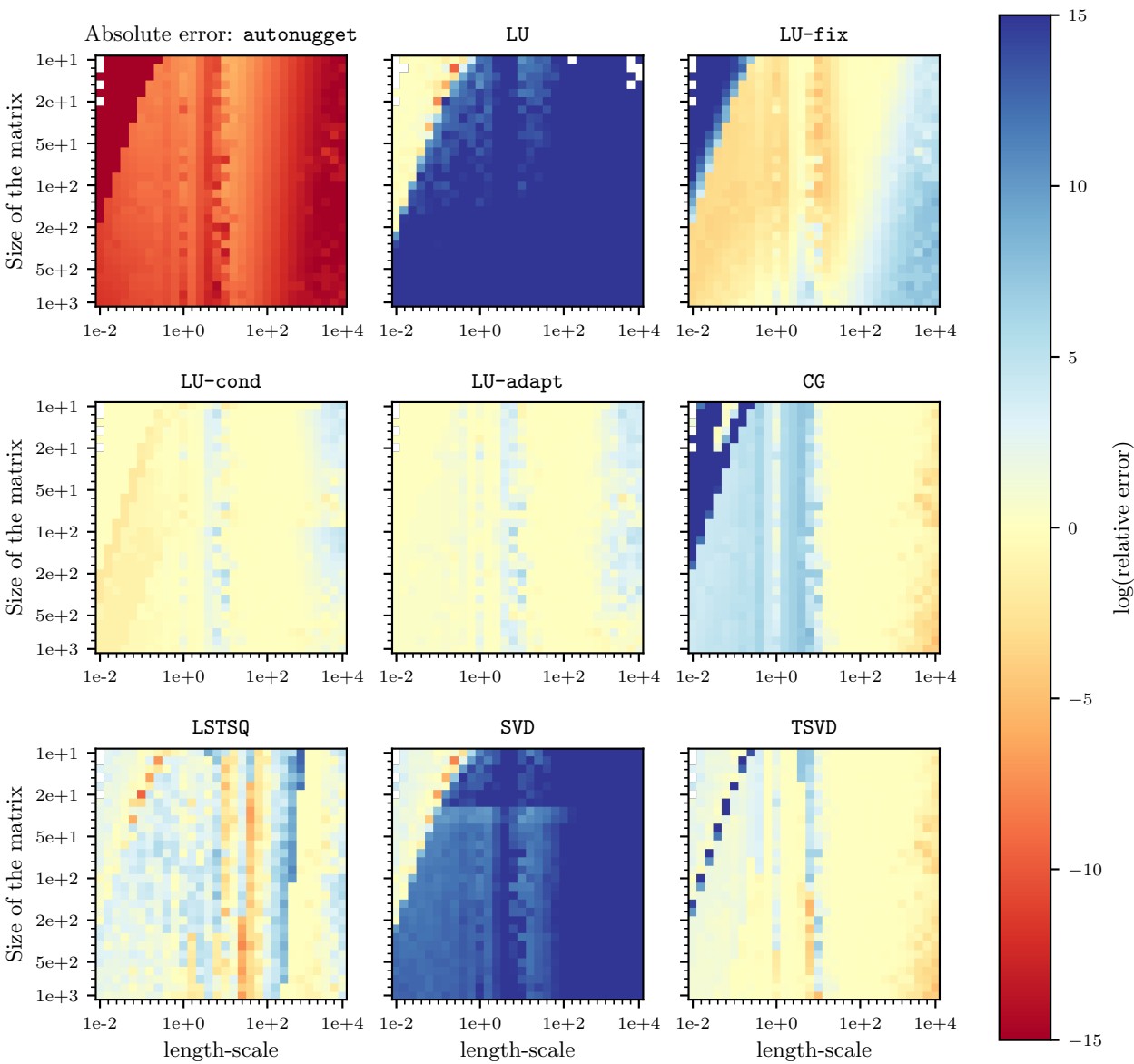

Figure 10: Fig. 3 computed with a different seed. This shows that the results are robust to stochasticity in the minimum eigenvalue computation.

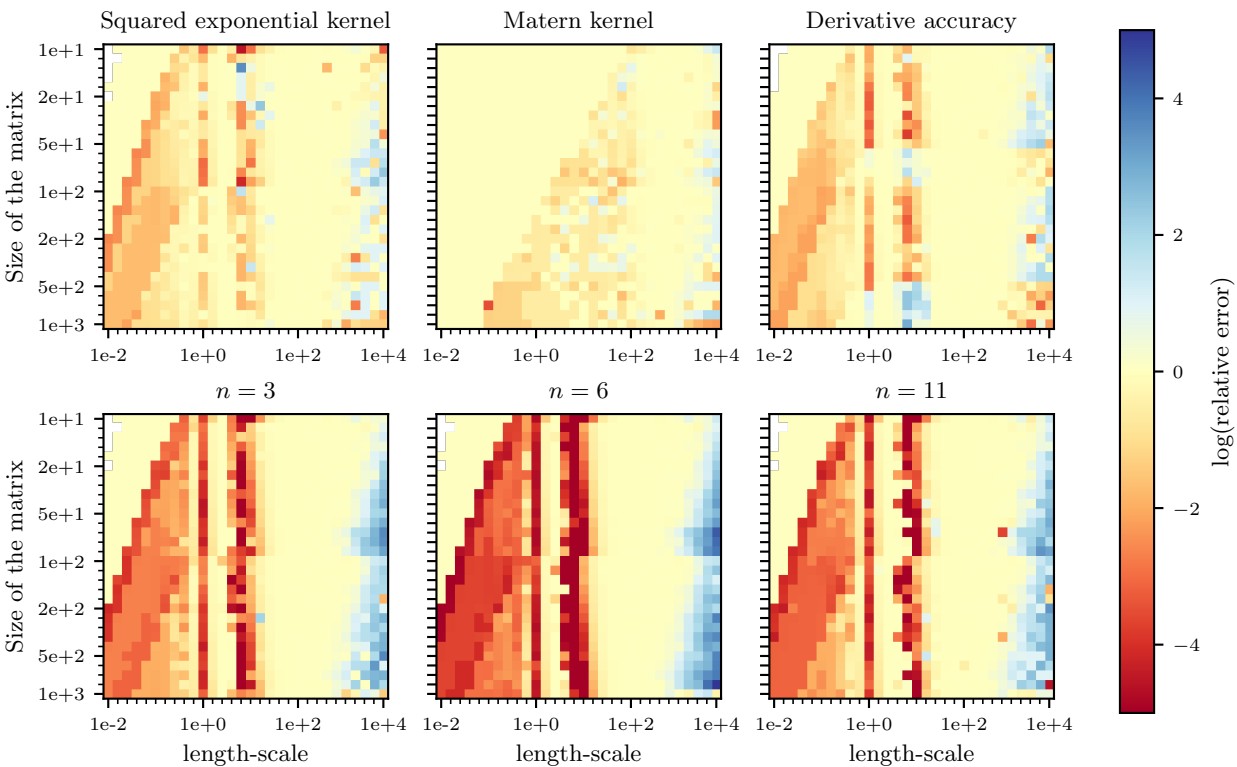

Figure 11: Accuracy assessment of `autonugget-cond`. Here we compared `autonugget-cond` to `autonugget`, in the settings described in Sections 4.2 and 4.3. The logarithm of the error relative to `autonugget` is reported.

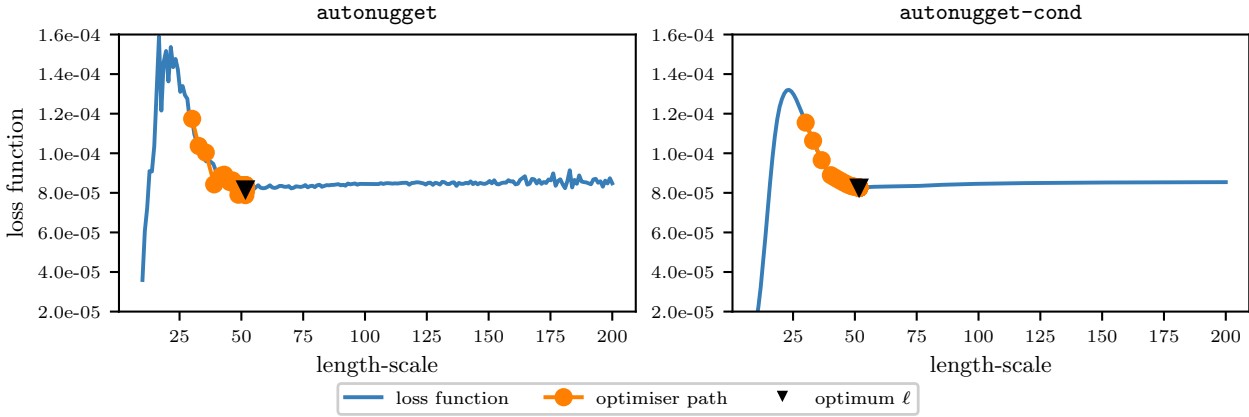

Figure 12: Comparison of `autonugget-cond` with `autonugget` for the problem in Section 4.4.

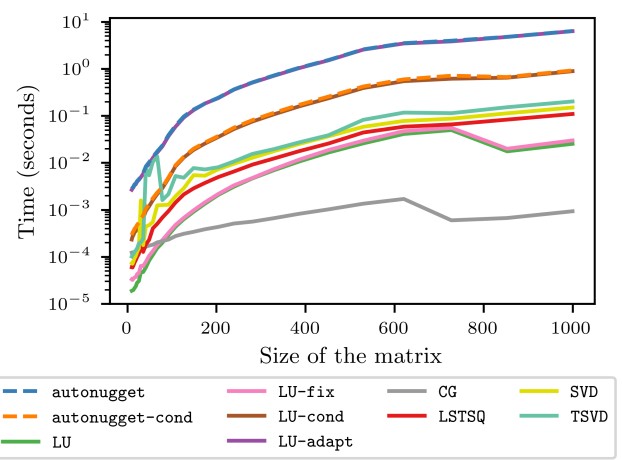

Figure 13: Comparison of wall time `autonugget` with other methods for the experiment in Section 4.2.

