# OpenReview forum: "Extrapolating from Regularised Solutions for Solving Ill-Conditioned Linear Systems in Machine Learning"
_TMLR — Accepted by TMLR_

### Review · Reviewer_USWH · 2026-01-29

**Summary Of Contributions:**

Authors propose to solve $Ax = b$ with ill-conditioned symmetric positive definite matrix $A$ using extrapolation. The recommended approach is as follows:
1. Solve $(A + \sigma I) x(\sigma) = b$ for several specifically selected $\sigma > 0$.
2. For each component approximate $x(\sigma)$ by polynomial, i.e., $x_i(\sigma) = p_i(\sigma)$.
3. Estimate $x_i(0)$ as $p_i(0)$.

The algorithm is implemented in JAX along with a custom derivative rule.

Authors compare the proposed technique with several baselines on selected well and ill-conditioned problems demonstrating their method is more robust.

**Audience:**

Yes

**Audience Explanation:**

Authors consider a basic task of solving ill-conditioned linear systems that often appear in machine learning. In addition authors advocate for extrapolation that is currently underutilised in machine learning and may lead to advances in certain problems. Given that, the work may be of interest to the research community working on the intersection of machine learning and numerical analysis.

**Claims And Evidence:**

Yes

**Claims Explanation:**

Authors provide both theoretical and numerical evidence that their method works. I have some doubts regarding both theory and empirical evaluation (explained below), but in my view the evidence authors provided are of sufficient quality.

**Requested Changes:**

**SVD**

The method of authors require:
1. Solution of $n\geq2$ linear systems for $d\times d$ dense matrices
2. Estimation of $\lambda_{\min}$ by $k$ matrix-vector products

Asymptotically, this imply we have $O(d^3)$ FLOPs and $O(d^2)$ memory, so we might as well perform SVD of matrix $A$ which gives us $\lambda_{\min}$, $\lambda_{\max}$, numerical rank, null space and solution to $(A+ I \sigma) x = b$ for all $\sigma$.

It seems that SVD is not seriously considered as a baseline besides LSTSQ that may call SVD internally.

Several questions are in order:
1. Can the authors please comment on why they do not think SVD is a valuable option?
2. Can authors try SVD supplemented with some adaptive threshold strategy and report the results?
3. Is it not easier to apply extrapolation after SVD is computed?

**Questions on numerical results**

1. The only absolute error reported is in Figure 8 in the appendix. It is hard to judge whether the results are good or not based mostly on the relative error.

   If for method $X$ absolute error is $10^{2}$ (bad), and for method $Y$ absolute error is $10^4$ (even worse), the logarithm of relative error $\log_{10}\left(10^{4} \big/ 10^{2}\right) = 2$ indicate that method $X$ is much better (blue colour on the plot), which is true but both of the method are not great.

   Can the authors report absolute error of, say, `autonugget`  $n=2$ for each experiment where relative error is reported?

2. The data present in Figure 8 is unclear.

   a. The third plot in the first row $n=1$ suggests that the results are given relative to the `autonugget` with no extrapolation. Yet the second plot in the first row has the title "no extrapolation". What does it mean?

   b. I assume the results in the second row are given relative to the no extrapolation `autonugget`. For well-conditioned systems methods with extrapolation seem to perform worse (blue patches in the middle of the plot). Why is that?

   c. I assume the results in the second row are given relative to the no extrapolation `autonugget`. With the increase of $n$ the relative error does not seem to improve significantly for ill-conditioned systems. Can the authors explain why this happens? Specifically, I am interested in how these results align with presented theory.

   d. The first plot in the first row that gives absolute error seems to indicate that for ill-conditioner systems the absolute error is already $\simeq 10^{-15}$. (i) What kind of method is that? Does it have extrapolation? (ii) If error is already about machine precision, does it mean that the improvement in accuracy is irrelevant?

3. For Gaussian Processes authors consider a tiny problem with the $25\times 25$ matrix. Can the authors comment on the scalability of their approach? It would be also great to see the comparison of wall-clock time needed by the `autonugget` and other standard approaches.

**Minor errors**

1. page 6

    Authors write that for spd $M$ matrix
    $$
    \lambda\_{\min}(M)^{-1} = \left\\|M^{-1}\right\\|\_{\text{op}} = \left\\|M\right\\|\_{\text{op}}^{-1} = \sup\_{\left\\|v\right\\|\_{2} = 1} \left\\|M v\right\\|\_2^{-1},
    $$
    where
    $$
    \left\\|M\right\\|\_{\text{op}} = \sup\_{\left\\|v\right\\|\_{2} = 1} \left\\|M v\right\\|\_2.
    $$
    First the equality imply
    $$
    \lambda\_{\min}(M)^{-1} = \left\\|M\right\\|\_{\text{op}}^{-1} \Rightarrow\lambda\_{\min}(M) = \left\\|M\right\\|\_{\text{op}},
    $$
    which is not the case since for spd matrix $\left\\|M\right\\|\_{\text{op}} = \lambda\_{\max}(M)$.

    Second
    $$
    \left\\|M\right\\|\_{\text{op}}^{-1} \neq \sup_{\left\\|v\right\\|\_{2} = 1} \left\\|M v\right\\|\_2^{-1}
    $$
    but rather
    $$
    \left\\|M\right\\|\_{\text{op}}^{-1} = \left(\sup\_{\left\\|v\right\\|\_{2} = 1} \left\\|M v\right\\|\right)^{-1}.
    $$
    The correct justification would be
    $$
    \sup\_{\left\\|v\right\\|\_{2} = 1} \left\\|M v\right\\|\_2^{-1} = \sup\_{\left\\|w\right\\|\_2 = 1} \frac{1}{\sqrt{\sum\_{i=1}^{N}\lambda\_i w\_{i}^2}} =  \frac{1}{\inf\_{\left\\|w\right\\|\_2 = 1}\sqrt{\sum\_{i=1}^{N}\lambda\_i w\_{i}^2}} = \frac{1}{\lambda\_{\min}(M)}
    $$

2. page 2 "‘nuisance’ parameter that one would ideally want to be small"

    I think this description is not accurate. As I understand, in statistics a nuisance parameter is any parameter that is not observed directly. For example, there can be unknown variance of noise. It can be estimated or integrated out in Bayesian methods.

---

> ### Author Response · Authors · 2026-04-20
>
> Thank you for the review and helpful comments.
>
> ### Regarding use of SVD:
>
> Our choice of using the default `numpy` solver was mainly due to numerical stability. Mathematically, performing a single SVD would give us all the quantities needed for all $\sigma$ in absolute precision, but in practice we found that this approach was sensitive to floating point precision arithmetic. Thus we resort to separate linear solves, whose error can be quantified as in Section 2.3.
> This approach could still be practical with further development and analysis of the error structure. Moreover, this leads to an interesting avenue of extrapolating in a space spanned by the SVD vectors as opposed to the current setup, that extrapolates in the standard Euclidean basis.
> Both of these avenues are interesting, but we will leave them for future work.
>
> We agree that both SVD and truncated SVD (with truncation at $10^{-8}$) are natural additional baselines, and have added these in Figure 3.
> The performance of truncated SVD is very competitive with the extrapolation approach, though there are still some regimes in which autonugget delivers improved performance.
> When computing derivatives, however, we found that the `jax` implementation of SVD is prone to failure, with this occurring particularly frequently for poorly conditioned matrices.
> We therefore still feel that our approach offers distinct advantages over a truncated SVD.
>
> ### About the numerical results:
>
> Thank you for your suggestion about reporting absolute errors. All the plots in the updated version now have plots of absolute error of the method being compared to.
> We have also included plots showing the absolute error of every method, in addition to the relative error plots.
>
> > The third plot in the first row $n = 1$ suggests that the results are given relative to the autonugget with no extrapolation. Yet the second plot in the first row has the title "no extrapolation". What does it mean?
>
> The title of the plots should have been '$m$' instead of '$n$', where $m$ is the degree of polynomial used for extrapolation, i.e., $m = n - 1$.
> Thus the third plot in the first is extrapolation using the first degree polynomial while the second plot in the first row is the one with no extrapolation. This has been corrected in the updated version.
>
> > I assume the results in the second row are given relative to the no extrapolation autonugget. For well-conditioned systems methods with extrapolation seem to perform worse (blue patches in the middle of the plot). Why is that?
>
> The results are relative to the default autonugget setup, which is extrapolating using a first degree polynomial. Thus the method without extrapolation (in the second plot of the first row) performs worse (for ill-conditioned systems and some not-so ill-conditioned systems) as compared to the method with extrapolation (in the third plot of the first row).
>
> > The first plot in the first row that gives absolute error seems to indicate that for ill-conditioned systems the absolute error is already $\approxeq 10^{-15}$. (i) What kind of method is that? Does it have extrapolation? (ii) If error is already about machine precision, does it mean that the improvement in accuracy is irrelevant?
>
> The first plot is the first row the absolute error of the default autonugget setup, which is extrapolating using a first degree polynomial. We agree that error here is already very low, and thus the additional improvement given by using a higher degree polynomial (as seen in the second row) is not very helpful. We thus recommend using just a first degree polynomial.

---

> ### Author Response · Authors · 2026-04-20
> **Response cont.**
>
> ### About the numerical results:
>
> > I assume the results in the second row are given relative to the no extrapolation autonugget. With the increase of the $n$ relative error does not seem to improve significantly for ill-conditioned systems. Can the authors explain why this happens? Specifically, I am interested in how these results align with presented theory.
>
> The second row is relative to autonugget with extrapolation using a first degree polynomial. With the increase of $m$ in the second row, the error seems to decrease for highly ill-conditioned systems (at the right end of the plots). However, more interestingly, the error does slightly increase for well-conditioned systems in middle of the plots. Theorem 1 says that using a higher degree polynomial would lead to lower error in $\infty$-norm in absolute precision. We report $2$-norm error in all our plots as this is generally more practically meaningful, and these also include the numerical errors.
> The apparently contradictory results is due to this mismatch of norms.
> While the theory says that there should be a strict improvement in the infinity norm, the $2$-norm can still fluctuate.
>
> From these empirical results, we recommend using a lower degree polynomial for extrapolation, especially since the additional improvement from a higher degree polynomial for ill-conditioned systems are not extremely useful as the first degree polynomial gives very small-error already (first plot of the first row).
>
> We realise that the title of Figure 8 is incorrect and the explanation is unclear. We have corrected this and added the additional explanation from above in the updated version.
>
> > Can the authors comment on the scalability of their approach? It would be also great to see the comparison of wall-clock time needed by the autonugget and other standard approaches.
>
> In terms of computational order, our method has the same scaling as solving the linear system (i.e., it scales cubically with dimension). However, in terms of FLOPS, our method involves solving additional linear systems (for the extrapolation) and performing additional SVDs (to identify the nugget).
> Therefore, it does have a higher cost, but it is higher by a constant factor.
> For our recommended settings (extrapolation with a linear polynomial), only one additional linear solve is required, and so the additional cost is dominated by the search procedure used to identify a nugget.
>
> To verify this practically, we have added a plot of wall-clock times in the appendix (Figure 13), which indeed shows that our method is a constant factor more expensive than the majority of competitors.
> We emphasise that our method has not been fully optimised for speed, so there are likely significant gains to be made here. Furthermore, the goal of our method is to provide a solver that is (possibly slow but) reliable, tuning-free and compatible with JAX, for prototyping.
>
> ### Regarding Minor Errors:
>
> Thank you for pointing out the error in the approximation of the minimum eigenvalue. We have corrected this in the updated version.
>
> > I think this description is not accurate. As I understand, in statistics a nuisance parameter is any parameter that is not observed directly.
>
> Apologies for this unclear terminology. We have changed this to 'numerical' parameter in the updated version.

---

> > ### Comment · Reviewer_USWH · 2026-05-01
> >
> > I would like to thank the authors for providing the absolute error, correcting the theoretical parts, and reporting the wall-clock time. In general, the authors' reply resolves my main concerns.
> >
> > I agree with reviewer EXXC that the authors should be more careful in framing the problem and their contributions. A good example of related work that also aims at finding an unregularized solution is https://arxiv.org/abs/1911.09988, where the authors stabilize the solution of $Ax \simeq b$ with the Vandermonde matrix $A$ using the Arnoldi method.
> >
> > Since the authors' contribution is more on the machine learning side, the discussion of alternative approaches becomes more nuanced. For example, the $\sigma$ that appears in Tikhonov regularization has a counterpart in Bayesian methods. In Bayesian methods, one has a distinct set of tools for selecting $\sigma$ (e.g., comparing evidence, marginalization with posterior distributions), which are suitable for handling “real data” with noise. It is not entirely clear to me how the approach advocated by the authors should be compared to the one used in Bayesian statistics. The two methods seem to solve two distinct problems.
> >
> > Similarly, Tikhonov regularization is often applied in cases where there is no unique solution at all, so it cannot be compared directly with Autonugget.
> >
> > I think an extended discussion on how $\sigma$ is selected in different contexts (regularization, Bayesian statistics) could strengthen the paper.

---

> > > ### Author Response · Authors · 2026-05-08
> > >
> > > Thank you for your suggestion. We agree that a discussion on how $\sigma$ is selected in different contexts would be a good addition to the paper, and are happy to add this discussion.

---

### Review · Reviewer_EXXC · 2026-02-05

**Summary Of Contributions:**

The paper presents "autonugget", a JAX package intended as a drop-in replacement for a direct linear solver when the system (Ax=b) is ill-conditioned. The method performs several Tikhonov-regularised solves with different nuggets (\sigma_i) and combines them through Richardson extrapolation to approximate the limit (\sigma->0). A custom JAX differentiation rule based on the implicit formula in Eq. (12)
is provided so that the routine can be used inside gradient-based training pipelines. Empirical demonstrations are given on kernel Gram matrices and a Gaussian-process hyper-parameter task.

---------------
Strength

- The work offers a "practical software tool" that is easy to use and claims "compatibility with automatic differentiation in JAX*", which may be convenient for rapid prototyping in ML workflows.

---------------
Main concerns

-The central problem, selection of the Tikhonov parameter, is a long-studied topic with established methods (GCV, discrepancy principle, SURE, TSVD, iterative regularisation). The manuscript presents it as an open challenge without engaging with this mature literature, so the scientific novelty appears limited.

-The method aims to recover the "unregularised limit" (\sigma->0), while in inverse problems and GP regression the "nugget" usually reflects noise or model mismatch. Extrapolating it away can contradict the statistical purpose of regularisation.

-Circular evaluation: Accuracy in Figure 3 is measured "relative to autonugget itself", not to ground truth or predictive risk, which favours the proposed method by construction.

-Comparisons rely on weak generic solvers (fixed nuggets, default CG, numpy lstsq) and omit standard alternatives such as GCV-chosen λ, TSVD, pivoted Cholesky, or early-stopped iterative methods.

-The need for AD is not fully justified; finite-difference or implicit approaches are viable and often simpler, while the extrapolation step may amplify variance and cost.

-The approach requires several direct linear solves, each with cubic cost; using n nuggets leads to O(n*d^3). This limits applicability to small matrices. For the medium–large dimensions common in ML prototyping, iterative or structured methods are typically preferred, so the practical scope of the method appears narrower than suggested.

-The method introduces several hyperparameters, number of extrapolation points n, reference design \Sigma, and polynomial degree m—with default choices but no strong justification.

-Additionally, it is limited to "symmetric positive definite matrices", excluding many prototyping problems with non-symmetric or indefinite systems


Overall, the paper reads primarily as a "software convenience contribution" rather than a substantive advance in regularisation or numerical linear algebra.

**Audience:**

No

**Audience Explanation:**

The problem addressed is old, long-studied and already handled by numerous well-established libraries and techniques. The proposed software offers little beyond convenience, and given the abundance of existing tools, it is unlikely to be adopted or of interest to the TMLR audience.

**Broader Impact Concerns:**

There are no significant ethical or societal concerns associated with this work.

**Claims And Evidence:**

No

**Claims Explanation:**

Accuracy and convergence are measured relative to the authors’ own method, creating a circular evaluation. Baselines are weak and omit standard λ-selection techniques. The GP experiment uses noiseless data, which is unrealistic and biases results toward their extrapolation approach. While the software works as intended, the evidence does not convincingly demonstrate general correctness, stability, or practical superiority across realistic settings.

**Requested Changes:**

Respectfully, I have no changes to suggest that would make this work suitable for publication. The topic is too classical, the proposed method offers only minor software convenience, and the evaluation is limited and circular. I would advise reconsidering or retiring the submission, as no adjustments would address the core issues of novelty, significance, or relevance to TMLR’s audience.

---

> ### Author Response · Authors · 2026-04-20
>
> Thank you for the review. We hope we have addressed the main concerns you have raised below.
>
> > - The central problem, selection of the Tikhonov parameter, is a long-studied topic with established methods (GCV, discrepancy principle, SURE, TSVD, iterative regularisation). The manuscript presents it as an open challenge without engaging with this mature literature, so the scientific novelty appears limited.
>
> Selection of the Tiknonov parameter is not the objective of our paper.
> Rather, our paper seeks to use the solution of Tikhonov regularised systems to determine, by extrapolation, the solution of the unregularised problem.
> We do not believe our paper argues that the Tikhonov problem is an open challenge, but that general solution of ill-conditioned linear systems is an open challenge, and we believe this latter statement is accurate.
>
> Of the methods you mention, employing the discrepancy principle or using iterative regularisation would not be suitable to for this problem as they assume regularisation as a goal. Additionally, the data-centric methods like generalised cross-validation (GCV) and Stein’s unbiased risk estimator (SURE) are more general regularisation methods, which again assume regularisation is of intrinsic interest which is not the setting of the paper.
>
> We agree that SVD and truncated SVD are relevant, and have now added SVD and TSVD as additional baselines in the updated version of the paper.
>
> > - The method aims to recover the "unregularised limit" (\sigma->0), while in inverse problems and GP regression the "nugget" usually reflects noise or model mismatch. Extrapolating it away can contradict the statistical purpose of regularisation.
>
> There are many settings where the unregularised limit is the genuine object of interest.
> For example, in Bayesian optimisation, emulation and probabilistic numerics, it can be quite unnatural to assume observational noise.
> In these settings the quantities one uses to build a regression model are evaluations of a computer model or a deterministic equation, and are generally not random.
> In the revised draft we have added more justification for this perspective in Section 1.
>
> > - Circular evaluation: Accuracy in Figure 3 is measured "relative to autonugget itself", not to ground truth or predictive risk, which favours the proposed method by construction.
>
> We apologise for the misunderstanding here. Throughout our work, the error is always computed relative to the ground truth. A heat map of the raw errors is difficult to interpret, because the tasks span a considerable range of difficulties.
>
> In contrast, our relative error is a fair and well-interpretable comparison metric, as positive relative error implies lower error in autonugget compared to the other method, allowing one to establish whether autonugget outperforms a competitor.
> We do not believe that it favours our method by construction; our plots still make clear in which cases autonugget performs poorly relative to competitors, since we used diverging colour maps to highlight areas where the relative error was negative.
> However, we agree that this does not give a complete picture of the error structure in some cases.
>
> We have now also added absolute error versions of each plot in the paper which used relative error as its metric in Figure 6. We hope that you will find the revised plots give a more complete picture of the performance of our method.
>
> > - Comparisons rely on weak generic solvers (fixed nuggets, default CG, numpy lstsq) and omit standard alternatives such as GCV-chosen λ, TSVD, pivoted Cholesky, or early-stopped iterative methods.
>
> GCV and early stopping each assume the regularised system to be the object of interest, which is not the setting of our paper.
> We have now added SVD and TSVD as additional baselines.
>
> > - The need for AD is not fully justified; finite-difference or implicit approaches are viable and often simpler, while the extrapolation step may amplify variance and cost.
>
> While we believe auto-diff compatibility is expected and underpins the success of modern ML, we acknowledge that there may be settings where the other approaches you have mentioned might have superior performance.
> However, we do not believe it is categorical.
> For example, finite difference approaches tend to scale poorly with parameter dimension and introduce numerical error. Additionally, while they do sometimes work well, but they are also prone to failing, and manually validating each instance of a finite-difference approximation presents a barrier to rapid prototyping. We are not aware of a straightforward way to construct an implicit differentiation rule for the extrapolated estimate, since the solution is not defined as the root of a single fixed-point equation.
> AD compatibility is not the central contribution of our paper. It is still a desirable property of the method, ensuring that gradients of the extrapolated estimate are exact and obtainable at negligible additional cost.

---

> ### Author Response · Authors · 2026-04-20
> **Response cont.**
>
> > - The approach requires several direct linear solves, each with cubic cost; using n nuggets leads to O(n*d^3). This limits applicability to small matrices. For the medium–large dimensions common in ML prototyping, iterative or structured methods are typically preferred, so the practical scope of the method appears narrower than suggested.
>
> We agree that our method incurs higher computational cost than other approaches, but we feel this is justified by the intended use as a prototyping tool, as well as the fact that options for solving nearly-singular systems are rather limited. We have nevertheless added this as a limitation in the updated version, and also present wall-clock times in the appendix.
> On this point we would note that, while autonugget is indeed slower, it appears to be by a constant factor.
> Moreover, our current implementation is not fully optimised, and it is likely that significant gains could be made by optimising the code.
>
> > - The method introduces several hyperparameters, number of extrapolation points n, reference design \Sigma, and polynomial degree m—with default choices but no strong justification.
>
> While our method introduces several hyperparameters, we justify our choices theoretically where possible, and where not possible, empirically. We choose a reference design known to be well-suited to polynomial extrapolation (as described in Section 3). The number of extrapolation points (and thus the degree of the polynomial) is chosen to be $2$ as the empirical results show that improved accuracy from additional points was not substantial enough to merit the additional computational cost. This is explained more clearly with respect Figure 9 in the updated version.
>
>
> > - Additionally, it is limited to "symmetric positive definite matrices", excluding many prototyping problems with non-symmetric or indefinite systems
>
> We agree that this is a limitation of our method. Expanding to non-SPD systems would certainly be of interest, but is not possible in our current theoretical framework, so we would defer such an expansion to future work.
> We have mentioned this in the conclusion.
> However, narrowing attention to SPD systems still makes the method applicable to a very large class of problems across machine learning and statistics.
>
> To summarise, we have added absolute error plots and additional baselines in the updated version. Our method focuses on solving unregularised linear systems, which we argue is still relevant in many ML applications. Our experiments thus mimic such settings, i.e., we test a noiseless data setting.

---

### Review · Reviewer_zMum · 2026-04-07

**Summary Of Contributions:**

The paper applies Richardson extrapolation to solve ill-conditioned symmetric positive definite linear systems $Ax=b$. It solves $N$ Tikhonov-regularized systems $(A+\sigma_i I)x_{\sigma_i}=b$, with $\sigma_i=h \sigma_i^{\mathrm{ref}}$, where $h$ is chosen automatically to balance extrapolation and numerical errors; it then extrapolates the map $\sigma \mapsto x_\sigma$ to $\sigma=0$. The reference points $\sigma_i^{\text{ref}}$ are taken from the geometric grid $\{2^{-j}\}_{0 \leq j \leq m}$, where $m$ is user-specified (defaulting to $m=1$), a choice motivated by the suitability of geometric grids for polynomial extrapolation. The method is released as a JAX-compatible Python package for rapid prototyping and is designed as a one-line replacement for standard solvers. The paper supports the method with upper bounds on both extrapolation and propagated numerical errors, which justify the automatic choice of $h$.

**Audience:**

Yes

**Audience Explanation:**

The paper addresses a practical gap that is likely to interest parts of the TMLR audience, especially researchers in optimization and numerical linear algebra who work with ill-conditioned systems, as well as ML researchers interested in stable and differentiable training pipelines.

**Broader Impact Concerns:**

No significant ethical concerns identified. The work addresses a numerical methods problem with no potential harmful applications.

**Claims And Evidence:**

Yes

**Claims Explanation:**

The paper is well-structured and easy to follow. The Python package is publicly available, and the theoretical error bounds are formally stated and proven. Empirical evidence is present but could be strengthened (see below).

**Requested Changes:**

- I would like the authors to include a table with explicit numerical values (mean error and standard deviation across multiple random seeds) for each baseline and each conditioning regime, to allow for precise quantitative comparison beyond the heatmap visualisations currently provided.
- I would like the authors to include wall-clock time comparison across methods, as this is central to the paper's rapid prototyping motivation and to support their claim that "the overall computational cost of autonugget is of the same order as a single application of a direct method."
- I am not fully sure about this, but the range of matrices the method is evaluated on feels a bit limited (essentially kernel/Gram matrices). Can the authors propose to evaluate on different matrices (maybe NTK matrices?)?

---

> ### Author Response · Authors · 2026-04-20
>
> Thank you for the review and helpful comments.
>
> > I would like the authors to include a table with explicit numerical values (mean error and standard deviation across multiple random seeds) for each baseline and each conditioning regime, to allow for precise quantitative comparison beyond the heatmap visualisations currently provided.
>
> Thank you for this suggestion, we have now added a table containing explicit numerical values (mean error and standard deviation across multiple random seeds) in the appendix Table 1 and Table 2. Additionally, we have added absolute error plots to each of our Figures.
>
> > I would like the authors to include wall-clock time comparison across methods, as this is central to the paper's rapid prototyping motivation and to support their claim that "the overall computational cost of autonugget is of the same order as a single application of a direct method."
>
> In terms of computational order, our method has the same scaling as solving the linear system (i.e., it scales cubically with dimension). However, in terms of FLOPS, our method involves solving additional linear systems (for the extrapolation) and performing additional SVDs (to identify the nugget).
> Therefore, it does have a higher cost, but it is higher by a constant factor.
> For our recommended settings (extrapolation with a linear polynomial), only one additional linear solve is required, and so the additional cost is dominated by the search procedure used to identify a nugget.
>
> We have added a plot of wall-clock times in the appendix (Figure 13), which indeed shows that our method is a constant factor more expensive than the majority of competitors.
> We emphasise that our method has not been fully optimised for speed, so there are likely significant gains to be made here. Furthermore, the goal of our method is to provide a solver that is (possibly slow but) reliable, tuning-free and compatible with JAX, for prototyping. We have also added this as a limitation of our approach in the conclusion.
>
> > I am not fully sure about this, but the range of matrices the method is evaluated on feels a bit limited (essentially kernel/Gram matrices). Can the authors propose to evaluate on different matrices (maybe NTK matrices?)?
>
> We evaluate autonugget on a range of non-kernel type symmetric positive definite matrices in Figure 7.
>
> We agree that neural tangent kernels is an interesting avenue to explore for our methodology, but we do not believe this is feasible during the review window.
> Moreover, these matrices are typically singular (i.e. SPSD), which makes application of our theoretical results challenging.
> We have highlighted this as an avenue for future work in the conclusion.

---

> > ### Comment · Reviewer_zMum · 2026-05-04
> >
> > I thank the authors for providing the explicit numerical values, adding the absolute error plots, reporting the wall-clock time, and highlighting the results on non-kernel matrices (which appear to be in Figure 8). These updates altogether resolve my main concerns.
> >
> > I agree with Reviewer USWH that the paper could be further strengthened by discussing how $\sigma$ is selected across different contexts. Regarding NTKs, I agree that their singular (SPSD) nature falls outside the current framework, and exploring such matrices remains an interesting avenue to further strengthen the paper.

---

> > > ### Author Response · Authors · 2026-05-08
> > >
> > > Thank you for your response. We agree with you and reviewer USWH that a discussion on $\sigma$ selection in different contexts would be a good addition to the paper, and are happy to add this discussion.

---

### Decision · Action_Editor_quvg · 2026-06-03

**Recommendation:** Accept with minor revision

**Additional Comments:**

Rev `USWH` asks for one more discussion :
> The only remaining problem is on the methodological side: the authors seem to seek unregularized solutions to linear systems with symmetric positive definite but severely ill-conditioned matrices, but they compare them with methods suitable for symmetric positive semidefinite matrices. In the former case, regularization is needed for numerical stability, which is only seen on digital computers. In the latter case, the solution is not unique or does not exist at all without regularization. In my view, a thorough discussion of the differences between the two situations should be added to the article.

**Audience:**

Yes

**Audience Explanation:**

Rev `EXXC`:
> The work is technically sound after revisions, and the implementation plus empirical study is careful in parts. It may be of interest as a practical tool for solving ill-conditioned SPD systems in ML prototyping contexts.

Rev `zMum`:
> [The paper] addresses a practical gap for ML and optimization researchers who work with ill-conditioned systems and need stable, differentiable training pipelines. By providing a tuning-free, JAX-compatible Python package for rapid prototyping, the method is highly relevant to the TMLR audience


Rev `USWH`:
> authors advocate for extrapolation that is currently underutilised in machine learning and may lead to advances in certain problems. Given that, the work may be of interest to the research community working on the intersection of machine learning and numerical analysis.

**Claims And Evidence:**

Yes

**Claims Explanation:**

The paper proposes a method and a JAX package, `autonugget`, to solve ill-conditioned linear systems. The approach is based on solving a sequence of Tikhonov regularized linear systems with decreasing regularization strength $\sigma_k$, then on using Richardson extrapolation to compute the solution corresponding to $\sigma_\infty=0$.

Reviewers were initially concerned with the framing of the contribution, but two thorough rounds of modifications, including changing the paper's title, have lead to a satisfactory version.